



# Climate reconstructions based on GDGTs and pollen surface datasets from Mongolia and Siberia: Calibrations and applicability to extremely dry and cold environments

Lucas Dugerdil[1,2], Sébastien Joannin[2], Odile Peyron[2], Isabelle Jouffroy-Bapicot[3], Boris Vannière[3], Bazartseren Boldgiv[4], and Guillemette Ménot[1]

[1]Univ. Lyon, ENS de Lyon, Université Lyon 1, CNRS, UMR 5276 LGL-TPE, F-69364, Lyon, France
[2]Université de Montpellier, CNRS, IRD, EPHE, UMR 5554 ISEM, Montpellier, France
[3]Université Bourgogne Franche Comté, CNRS UMR 6249 Laboratoire Chrono-environnement, F-25030, Besançon, France
[4]Ecology Group, Department of Biology, School of Arts and Sciences, National University of Mongolia, Ulaanbaatar 14201, Mongolia

**Correspondence:** Lucas Dugerdil, lucas.dugerdil@ens-lyon.fr

**Abstract.** Our understanding of climate and vegetation changes throughout the Holocene is hampered by biases in the proxy representativeness in sedimentary archives. Such potential biases are identified by comparing proxies to modern environments.

Consequently, it becomes important to conduct multi-proxy studies and robust calibrations. The taiga-steppes of the Mongolian plateau, ranging from the extremely cold-dry Baikal basin to the Gobi desert, are characterized by low annual precipitation and continental annual air temperature as well as livestock grazing. The characterization of the climate system of this area is crucial for the understanding of Holocene Monsoon Oscillations. This study focuses on the calibration of proxy-climate relationships for pollen and glycerol dialkyl glycerol tetraethers (GDGTs) by comparing large published Eurasian calibrations

with a set of 53 new surface samples (moss, soil and surface sediments). We show that: (1) preserved pollen assemblages are clearly imprinted on the extremities of the ecosystem range but mitigated and unclear on the ecotones; (2) for both proxies, inferred relationships depend on the geographical range covered by the calibration database as well as on the sample nature; (3) local calibrations, even those derived to the low range of climate parameters encompassed in the study area, better reconstruct climatic parameters than the global ones by reducing the limits for saturation impact, and (4) a bias in climatic reconstructions

is induced by the over-parameterization of the models by addition of artificial correlation. We encourage the application of this surface calibration method to consolidate our understanding of the Holocene climate and environment variations.



# 1 Introduction

Since the understanding of the interactions between climate model outputs and their input proxies is a major issue in future
climate change modelling, resolving the issue of climate proxy calibration is crucial (Braconnot et al., 2012). Current changes
in extremely cold environments (Masson-Delmotte, 2018), such as Mongolia and Siberia, are amplified compared with other
places around the world (Tian et al., 2014) and the drivers of the current degradation of Mongolian environments still need
to be understood. From a climatic point of view, Mongolia is on a junction between the westerlies which are driven by the
North Atlantic Oscillation (NAO) and the East Asian Summer Monsoon which is linked to the El Niño-Southern Oscillations
(ENSO) and the Inter-tropical Convergence Zone. The Mongolian plateau is a hinge area: the high altitude of the Altaï range
to the west and the Sayan range to the north-west of the country block both the westerlies arriving from the northern Atlantic
ocean through the Siberian basin and the East Asian Summer Monsoon (EASM, Chen et al., 2009) (ITCZ, An et al., 2008).

Lake sediment archives are commonly used to infer past variations of the climate systems associated with vegetation and
human land use (Lehmkuhl et al., 2011; Felauer et al., 2012; Wang and Feng, 2013). Among the proxies available, pollen and
geochemical biomarkers are commonly used as past climate indicators (Ter Braak and Juggins, 1993; Weijers et al., 2007b).
The pollen signal displays shifts in vegetation composition and structure (Bennett and Willis, 2002) and allows quantitative
reconstructions of climate parameters such as precipitation regime and temperatures (Ter Braak and Juggins, 1993; Birks et al.,
2010; Ohlwein and Wahl, 2012; Wen et al., 2013; Cao et al., 2014). Since vegetation structure and pollen production are mainly
influenced by climatic parameters (Zheng et al., 2008), the palaeo pollen signal is very often interpreted as a response to the
climate variations through time (Kröpelin et al., 2008; Wagner et al., 2019). Even if human activities influence pollen rain as
well (Hjelle, 1997; Hellman et al., 2009a), these empirical observations of the pollen-climate relation leads to the development
of semi-quantitative (Ma et al., 2008) and quantitative calibrations (Brewer et al., 2007; Schäbitz et al., 2013) of the signal.
Different methods have been developed to reconstruct past climates: Indicator Taxa approaches, Assemblages approaches,
Transfer Functions (TF) and methods based on vegetation models (Birks et al., 2010; Bartlein et al., 2011; Ohlwein and Wahl,
2012). More precisely, these methods are: the inverse method (IM, Huntley and Prentice, 1988), the Weighted Averaging Partial
Least Squares regression (WAPLS, Ter Braak and Juggins, 1993; Ter Braak et al., 1993), the Artificial Neural Networks (ANN,
Peyron et al., 1998), the Modern Analogue Technique (MAT, Overpeck et al., 1985; Guiot, 1990; Jackson and Williams, 2004),
the Response Surface Technique (RST, Bartlein et al., 1986), Probability Density Functions (PDF, Kühl et al., 2002), Modified
Mutual Climate Range Method (MMCRM, Klotz et al., 2003, 2004), Bayesian Hierarchical Models (BHM, Ohlwein and Wahl,
2012), etc. Despite some problems and pitfalls, Seppä et al. (2004) demonstrated that pollen-inferred climate reconstructions
are generally consistent and agree well with other independent climatic reconstructions (for northern Europe).

Among the glycerol dialkyl glycerol tetraethers (GDGTs), four distinct groups have been described: the isoprenoid-GDGTs
(isoGDGTs, Hopmans et al., 2000), the branched-GDGTs (br–GDGTs, Damsté et al., 2000; Weijers et al., 2007a, b; Crampton-
Flood et al., 2019), the Glycerol Monoalkyl Glycerol Tetraethers (H-GDGTs, Naafs et al., 2018) and the hydroxy-GDGTs





(OH-GDGTs, Liu et al., 2012). GDGT assemblages reflect bacterial activity in rivers (De Jonge et al., 2014b), soil (De Jonge et al., 2014a) or lakes (Dang et al., 2018) which is also linked to climate parameters (Weijers et al., 2007b), soil typology and vegetation cover (Davtian et al., 2016), which in turn imply land cover and land use. Accurate determinations of the relation-

ships between GDGT assemblages and climate still need some improvements (Naafs et al., 2018) and especially on local to regional scales and in extreme environments.

To evaluate the provenance and the climatic information br–GDGTs bear, several indices have been proposed in the literature (Table. S1). The first proposed to determine the origin of the GDGTs is the BIT index (Hopmans et al., 2004), followed by the

$III_a/II_a$ index Xiao et al. (2016). Furthermore, it has been shown both empirically (Weijers et al., 2004; Huguet et al., 2013) and on cultures of pure strains (Salvador-Castel et al., 2019 in press) that organisms adjust their membrane plasticity by the degree and the position of these compounds. The br–GDGT assemblages are also a function of the bacterial species (Naafs et al., 2018) present in the environment. To monitor these changes, CBT and MBT indexes linked to environmental factors such as climate and soil parameters have been proposed (Weijers et al., 2007b; Huguet et al., 2013). Some more accurate indexes

have been proposed by De Jonge et al. (2014a) to limit the multi-correlation systems such as $MBT'_{5Me}$ which is independent of the pH and $CBT_{5Me}$ which is more representative of the soil pH than the former version of the index. The statistical relevance of these indexes is a major issue in br–GDGT calibration (Crampton-Flood et al., 2019). Some regional indexes for soil temperature such as $Index_1$ (De Jonge et al., 2014a) and $Index_2$ for Chinese soils (Wang et al., 2016) have been explored too. For the moisture variations, the $R_{i/b}$ index has been proposed as a reliable aridity proxy (Yang et al., 2014; Xie et al., 2012).

It has been shown that a linear relation exists between these GDGT indexes and some climatic features (Yang et al., 2014; Lei et al., 2016). The Siberian and Mongolian surface soil samples are used to calibrate new climatic relation with GDGTs.

In recent decades, climate calibrations have been proposed in arid central Asia from pollen semi-quantitative climate reconstruction (Ma et al., 2008), pollen transfer functions (Herzschuh et al., 2003, 2004; Cao et al., 2014; Zheng et al., 2014), and

br–GDGT regression models (Sun et al., 2011; Yang et al., 2014; Ding et al., 2015; Wang et al., 2016; Thomas et al., 2017). These studies focus on areas around the EASM line (fig. 2, Chen et al., 2010; Li et al., 2018), while our study took place on the northernmost part of this climatic system (Haoran and Weihong, 2007). Moreover, we apply for the first time a multi-proxy calibration comparison study.

The aim of this study is to take advantage of new, modern surface sample datasets in Siberia and Mongolia to propose an adapted calibration of pollen and bacterial biomarker proxies for cold and dry environments. For that purpose, local calibrations are compared with large calibrations to infer the influence of calibration scale and proxy types on derived climatic parameters. Our approach is summarized in the following steps:

1. Collection of a new set of modern surface samples for Mongolia with homogeneous characterisation of their bioclimate

environment.



2. Evaluation of the match between actual bioclimate environments and the associated pollen rain and biomarker assemblages.

3. Creation of local Siberia-Mongolia climate calibrations for pollen and GDGTs and application of these calibrations to provide climate reconstructions.

4. Comparison of local and global calibrations on the Mongolian study case.

5. Discussion of the implications of the calibration discrepancies in terms of climatic reconstructions in arid and cold environments.

## 2  Mongolian and Siberian Study Area

### 2.1  Sampling Area and Sample Types

The study area lies from $52°29'$N to $43°34'$N in latitude and from $101°00'$E to $107°06'$E in longitude. The surface samples were collected around Lake Baikal (*n=12*, Irkoustsk Oblast, Russia, Fig. 1.G) and the samples gathered throughout Mongolia were taken in autumn 2016 following five transects *(n=29)*: in the Sayan range (*MRUT1*), in the Khentii mountain range (*MMNT1* and *MMNT2*, Fig. 1.E), in the Orkhon valley (*MMNT3*), in the Gobi desert and the Gobi-Altai range (*MMNT4*, Fig. 1.D). Finally, the Khangai mountains were sampled with surface and top lacustrine sediment samples (*MMNT5S01* to

*MMNT5M07*, Fig. 1.F). For each site, the *Garmin eTreX10* was used to record GPS coordinates to five-meter accuracy (Supplementary Table S1). Depending on aridity and vegetation at each site, a soil or a moss polster was sampled. Moss acts as a pollen trap recording a three to five-year mean pollen signal (Räsänen et al., 2004). In drier areas, the soil surface samples have the same function, in spite of a lower pollen conservation (Lebreton et al., 2010). In parallel, the two types of sample were also used for actual GDGT analysis following the calibration approaches presented in De Jonge et al. (2014a), Davtian et al. (2016)

and Naafs et al. (2017, 2018).

### 2.1.1  Vegetation and Biomes

The central part of the Mongolian plateau is characterized by a dry and cold flat desert with a 1220 m.a.s.l. median elevation (Fig. 1.A, Wesche et al., 2016) and is intersected in its northern part by the Khentii range and in its southern part by the Gobi-Altai range aligned along a NW-SE direction. A wet and cold highland in the Khangai ranges culminates at 4000 m.a.s.l

and a flatter and wetter Mongolian area, the Darkhad basin, is located in the north, close to the Russian border on the edge of the Siberian Sayan range. In the northernmost area, the geography is characterized by the Baikal lake basin at a lower altitude (around 600 m.a.s.l) (Fig. 1.G) Demske et al., 2005).

The distribution of vegetation and biomes follows a latitudinal belt organization: in the North the boreal forest presents

a mosaic of light-taiga dominated by *Pinus sylvestris* mixed with birches (Demske et al., 2005). On the Mongolian plateau,





**Figure 1. A**: Topographic map of Mongolia (from ASTER data) with the surface samples and weather stations considered in the present study; **B**: Mean Annual Precipitations; **C**: Mean Annual Air Temperatures; **D**: Focus on the samples surrounding the Taatsiin Tsagaan Lake, Gobi desert; **E**: Focus on the samples along a valley in the Khentii range; **F**: Localisation of Khangai surface samples; **G**: Focus on the Baikal Lake transect. The Mongolian GIS Data is issued from a dataset ASTER (https://biosurvey.ku.edu/directory/nicholas-kotlinski), the meteorological dataset from WorldClim2 and infrastructures from public dataset (ALAGaC) (https://marine.rutgers.edu/ cfree/gis-data/mongolia-gis-data/)





the dark-taiga dominated by larches (*Larix sibirica*) and Siberian pines (*Pinus sibirica*) also presents some spruces and fewer birches (*Betula* spp.). The Mongolian taiga is constrained to a region spanning from the Darkhad Basin to the Khentii range (Fig. 1.A). On the north face of the Khangai piedmont, the vegetation is dominated by a mosaic of forest-steppe ecosystems: the steppe is dominated by the *Artemisia* spp. associated with Poaceae, Amaranthaceae, Liliaceae, Fabaceae and Apiaceae.

On these open-lands there are some patches of taiga forest, following roughly the broadside and the northern face of the crest letting on to the grasslands in the valley. The two last vegetation layers through the elevation gradient is an alpine meadow dominated by Cyperaceae and Poaceae with a huge floristic biodiversity and an alpine shrubland with pioneer vegetation on the summits. On the southern slope of the range, the ecotone between the steppe and the desert vegetation extends hundreds of kilometers from the northern part of the Gobi desert (with water supplied by the Gobi lake area in the middle) to the Gobi-Altai

range in the south (Klinge and Sauer, 2019). In the southernmost part of the country, the warm and dry climate conditions favour desert vegetation dominated by Amaranthaceae, Nitrariaceae and Zygophyllaceae. The vegetation cover is lower than 25% and is mainly composed of short herbs, succulent plants and a few crawling shrubs.

## 2.2    Bioclimate Systems

In the central steppe-forest biome the vegetation is marked by an ecotone with short grassland controlled by grazing in the valley and larches on the slopes. The forest is gathered in patches constituting between 10% and 20% of the total vegetation cover. There are also some patches of *Salix* and *Betula* riparian forests among the sub-alpine meadows on the upper part of the range. This vegetation is characteristic of the northern border of the Palaearctic steppe biome (Wesche et al., 2016). This biome is characterised by a range of 800 to 1600 m.a.s.l, a Mean Annual Air Temperature (MAAT, Fig. 1.C) between -2 and

2°C and a Mean Annual Precipitations (MAP, fig. 1.B) from 180 to 400 $mm.yr^{-1}$ (Wesche et al. (2016) based on Hijmans et al., 2005). In Mongolia, even if the MAP are very low ($MAP_{Mongolia} \in [50; 500] mm.yr^{-1}$), the major part of the water available for plants is delivered during late spring and early summer, in contrast to Mediterranean and European steppes (Bone et al., 2015; Wesche et al., 2016). These seasons are the optimal plant growth periods. Mongolian summer precipitations are controlled by the East Asian Summer Monsoon system (EASM) instead of the Westerlies' winter precipitation stocked onto

the Sayan and Altai range (Fig. 2, An et al., 2008).

## 3    Methods

### 3.1    Pollen Analysis and Transfer Functions

Different chemical processes were performed on the samples: the mosses were deflocculated by $KOH$ and filtered by $250\mu m$

and $10\mu m$ sieves to eliminate the vegetation pieces and the clay particles. Then, acetolysis was performed to destroy biological cells and highlight the pollen grains. For the soil samples, 2 steps of $HCl$ and $HF$ acid attacks were added to the previous pro-





tocol to remove all the carbonate and silicate components. All the residuals were finally concentrated in glycerol and mounted between slide and lamella. The pollen counts were carried out with a *Leica DM1000 LED* microscope on a $40\times$ magnification lens. The total pollen count size was determined by the asymptotic behaviour of the rarefaction curve. This diagram was plotted

during the pollen count using *PolSais 2.0*, software developed in *Python 2.7* for this study. The rarefaction curve was fitted with a logarithmic regression analysis. The counter was suspended whenever the regression curve reached a flatter step (Birks et al., 1992). A threshold for the local derivation at $dx/dy = 0.05$ was set. The total count is usually around $n \in [350; 500]$ grains for steppe or forest and $n \in [250; 300]$ for desert slides.

Among all of the pollen-inferred climate methods, the MAT and the WAPLS were applied in this study on 4 different modern pollen datasets. The MAT consists of the selection of a limited number of analogue surface pollen assemblages with their associated climatic values. (Jackson and Williams, 2004); while the WAPLS uses a Weight Average correlation method on a limited number of Principal Components connecting the surface pollen fraction to the climate parameters associated (Ter Braak and Juggins, 1993; Ter Braak et al., 1993). The first dataset, called New Mongolian-Siberian Database (NMSDB),

is composed of pollen surface samples analysed in this study ($N = 49$, Fig. 2). The second one is the same subset aggregated to the larger Eurasian Pollen Dataset (EAPDB) compiled by Peyron et al. (2013, 2017). From this dataset of 3191 pollen sample sites, a pollen–*Plant Functional Type* method was applied to determine the biome for each sample according to the actual pollen rain (Prentice et al., 1996; Peyron et al., 1998). Then, only the Cold Steppe (COST) dominant samples were extracted from the main dataset and aggregated with the NMSDB to produce the COSTDB ($N = 482$ sites). Finally, a scale-intermediate dataset

of samples located within the Mongolian border merged with the Mongolian New dataset is presented as MDB ($N = 151$ sites). The relation between each taxa and climate parameter was checked and then the MAT and WAPLS methods were applied with the *Rioja* package from the R environment (Juggins and Juggins, 2019).

### 3.2 SIG Bioclimatic Data

Because Mongolia and Siberia have relatively few weather stations (Fig.1.A), climate parameters were extracted with R from

the extrapolated climatic database *WorldClim2* (Fick and Hijmans, 2017). We used Mean Annual Precipitation (MAP, Fig.1.B), Mean Annual Air Temperature (MAAT, Fig.1.C), as well as temperatures and precipitations for spring, summer and winter ($T_{spr}$, $P_{spr}$, $T_{sum}$, $P_{sum}$, $T_{win}$ and $P_{win}$), Mean Temperature of the Coldest Month (MTCO) and the Mean Temperature of the Warmest Month (MTWA) in this study to characterize the actual climate. The elevation data and the topographic map originate from the *ASTER* imagery (Fig.1.A). The biome type for each site derives from the *LandCover* database (Olson et al., 2001),

classification and field trip observations.

### 3.3 GDGT Analysis

After freeze drying, about 0.6 grams of surface samples were subsampled. The Total Lipid Extract (TLE) was microwave extracted (*MARS 6 CEM*) with dichloromethane (DCM):MeOH (3:1) and filtered on empty SPE cartridges. The extraction was processed twice. Following Huguet et al. (2006), $C_{46}$ GDGT (GDGT with two glycerol heed groups linked by $C_{20}$ alkyl chain





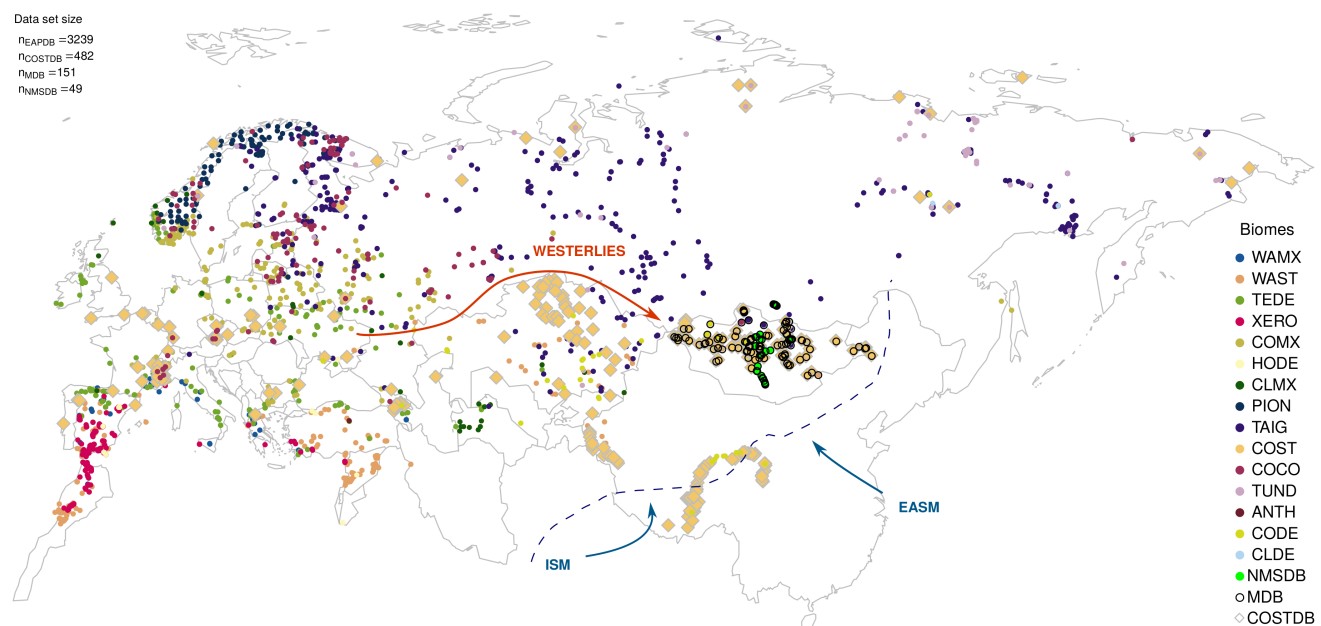

**Figure 2.** Eurasian map of all the pollen surface samples included in the database. The color code refers to the biome pollen inferred for each site. The codes are **WAMX**, Warm Mixed Forest; **WAST** Warm Steppe; **TEDE**, Temperate Deciduous Forest; **XERO**, Xerophytic Shrubland; **COMX**, Cool Mixed Forest; **HODE**, Hot Desert; **CLMX** Cold Mixed Forest; **PION** Pioneer forest; **TAIG**, Taiga forest; **COST**, Cold Steppe; **COCO**, Cold Conifer Forest; **TUND**, Tundra; **ANTH**, Anthropic environment; **CLDE**, Cold Deciduous Forest; **CODE**, Cold Desert. The thickest points underline the COST samples selected in this study for the transfer function method. The arrows indicate the main climatic system driving the Mongolian climate: in orange the Westerlies arriving from the North Atlantic ocean and in blue the Indian Summer Monsoon (ISM) and the East Asian Summer Monsoon (EASM). The dashed line represents the EASM limit following Chen et al. (2010), (Li et al., 2018) and Haoran and Weihong (2007) for the northernmost part of the boundary.

and two $C_{10}$ alkyl chains) was added as internal standard for GDGT quantification. Then, apolar and polar fractions were separated on an alumina SPE cartridge using hexane: DCM (1:1) and DCM/MeOH (1:1), respectively. Analyses were performed in hexane: iso-propanol (99.8:0.2) by High Performance Liquid Chromatography Mass Spectrometery (HPLC-APCI-MS, Agilent 1200) proceeded in LGLTPE-ENS de Lyon, Lyon, following Hopmans et al. (2016) and Davtian et al. (2018).

Statistical treatments on isoprenoid-GDGTs (Fig.4.A) and branched-GDGTs (Fig.4.B) abundances were treated following two methods presented in Deng et al. (2016), Wang et al. (2016) and Yang et al. (2019): compounds were gathered by chemical





structures as cycles (CBT) or methyl groups (MBT, De Jonge et al., 2014a) or compounds and especially the br–GDGTs were expressed as fractional abundance $[x_i]$ (Fig.4.B, Sinninghe Damsté, 2016), as follows:

$$f[x]_i = \frac{n_i}{\sum_{j=1}^{N_{br--GDGT}} x_j} \tag{1}$$

To infer temperature from br–GDGT abundances, two kinds of model were applied: the linear relation between temperature and MBT–CBT indexes, and a Multiple Regression (MR) model between one climate parameter and a proportion of multiple br–GDGT fractional abundances. For the simple linear regression model, a correlation matrix between climate parameters and indexes was calculated using the *corrplot* Rcran library. For the MR models, we developed in the R environment a *Stepwise Selection Model* (SSM, Yang et al., 2014) to determine the best fitting model connecting climate parameters with br–GDGTs

fractional abundances. Then we gathered some of the climate–GDGT linear relations established in previous papers (De Jonge et al., 2014a; Naafs et al., 2017, 2018; Sinninghe Damsté, 2016; Yang et al., 2014, 2019) focusing on a single climatic parameter, MAAT (Supplementary Table S2). These models were clustered into 3 categories: the type of sample, the geographical area and the statistical model. According to the type of environment from which the samples originated, there was peat, soil and lake-inferred modelling. For the geographical area of samples, we discriminated the regional model (on the country or district

scale) from the global model made on the world scale. Finally, there were 2 statistical families of model: the first one was built on common ratios like the MBT–CBT, and the second one was inferred with the multiple regression model. All these models were applied on the Mongolian surface samples and compared with the actual MAAT value at each site.

### 3.4    Statistical Analyses

GDGTs and pollen matrix were analysed with a Principal Components Analysis (PCA) to determine the axes explaining

the variance within the samples. The biotic values (pollen and GDGTs) were also compared to abiotic parameters (climate, elevation, location and soil features) by the way of a Redundancy Analysis (RDA). The regression models were run with the $p-value < 0.05$ for the relevance of the model, the $R^2$ for the level of correlation between the variables, the RMSE to determine the climate error of the models and Akaike's information criterion (AIC) to quantify the over-parameterization effect of multiple regression models (Arnold, 2010; Symonds and Moussalli, 2011). All the statistical analyses were performed with

the *Rcran* project, using the *ade4* package (Dray and Dufour, 2007) for multivariate analysis. All the plots were made with the *ggplot2* package (Wickham, 2016) or the *Rioja* package (Juggins and Juggins, 2019) for the stratigraphic plot and the pollen clustering using the CONISS analysis method (Grimm, 1987).



## 4 Results

### 4.1 Pollen, Climate and Ecosystems Relations

#### 4.1.1 Pollen Rain and Vegetation Representation

The pollen rain (Fig. 3) is dominated by six main pollen taxa: *Pinus sylvestris*, *Betula* spp., *Artemisia* spp., Poaceae, Cyperaceae and Amaranthaceae. The pollen diagram, sorted by bio-climate from the wet and relatively warm Siberian basin on the upper part to the dry-warm Gobi desert on the bottom, presents a net AP decrease from 85% to 5%. In Fig. 5.C we can see that 34.26 % of the variance is explained by $PC_1$ extending from positive values associated with NAP (Amaranthaceae, Poaceae and

*Artemisia* spp.) to negative values associated with AP (*Pinus* undet., *Betula* spp. *Picea obovata* and *Larix sibirica*). This trend shows the transition between ecosystems, marked by the seven main *CONISS* clusters (Fig. 3) and $PC_1$ and $PC_2$ variations (fig. 5.C). Below are the over-representative main *taxa* for each of the Siberian–Mongolian transect ecosystems:

1. **Light taiga** dominated by *Pinus sylvestris* ($> 70\%$), *Pinus sibirica* and very low NAP ($< 5\%$).

2. **Mixed dark taiga–birches sub-taiga** with an assemblage of *Larix sibirica*, *Picea obovata*, *P. sylvestris* and *P. sibirica*.

3. **Forest-steppe** ecotone same AP assemblages that the dark taiga ecosystem with 20% of *Artemisia* spp., plus occurence of Poaceae, Cyperaceae, *Thalictrum* spp. and *Convolvulus* spp.

4. **Steppe** still dominated by *Artemisia* spp. (30%) and rising Poaceae (25%), Brassicaceae

5. **Alpine meadow** overpowered by Cyperaceae up to 50 %, Poaceae, Brassicaceae, Amaranthaceae and *Convolvulus* spp.

6. **Steppe-desert** ecotone highlighting by the transition between Amaranthaceae–Caryophyllaceae community and Poaceae–

*Artemisia* spp. assemblages.

7. **Desert** dominated by Amaranthaceae (from 25 % to 65 %) and by rare pollen-type Caryophyllaceae, *Thalictrum* spp., *Nitraria* spp. and *Tribulus* spp.

#### 4.1.2 Pollen – Climate Interaction

The pollen rain trends follow similar variations to bio-climate parameters in MAP, MAAT and elevation (Fig. 3). Highest

AP/NAP values are correlated to large MAP (up to 500 $mm.yr^{-1}$) and relatively high MAAT (around 1 °C), in the low range Siberian basin. Then the transition between AP and NAP dominance is marked by decreases in both MAAT (-1 °C) and MAP (300 $mm.yr^{-1}$) connected to the high-elevation Khangai range. Finally, the dominance of NAP in the Gobi desert area is linked to very arid conditions (MAP $< 150mm.yr^{-1}$) and relatively warm MAAT (up to 4 °C). The correlation between the taxa themselves and climate parameters is $R^2 = 0.38$ (RDA, Fig.5.D). Rise in MAAT is associated with that of Amaranthaceae,

Poaceae, *Sedum*-type and Caryophyllaceae percentages. On the contrary, decrease in MAAT is associated with a rise in the AP





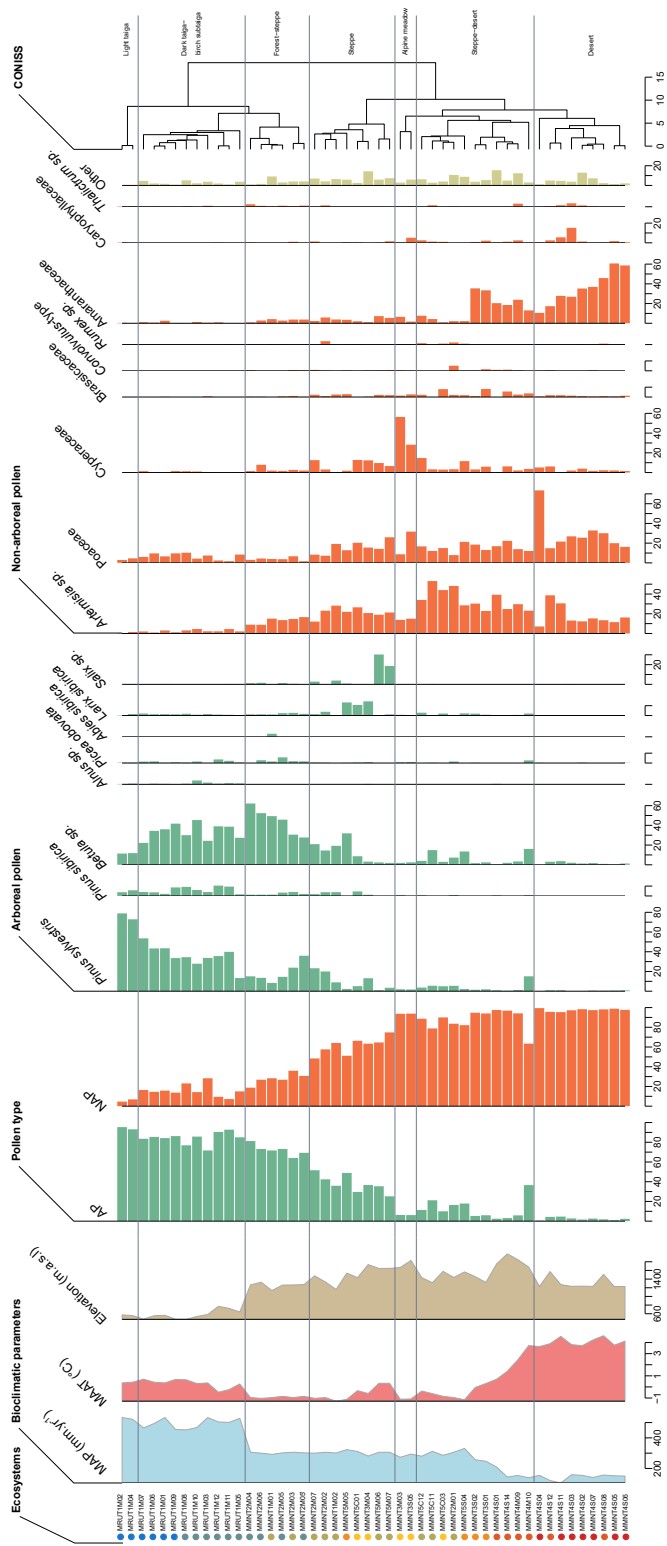

**Figure 3.** Simplified surface pollen diagram, bio-climatically sorted, of the Siberian–Mongolian transect. The pollen taxa are expressed in %TP. The Ecosystem Units were determined with a CONISS analysis. The left hand coloured dots represent the ecosystem for each sample from dark taiga (deep blue), light taiga, steppe-forest, alpine meadow, steppe, steppe-desert and desert (deep red). The color scale is presented in Fig. 5.





and Cyperaceae, *Artemisia* spp. and Brassicaceae percentages. MAP, fairly related to $RDA_1$, rises with AP and decreases with NAP (Fig.5.D). Finally, the elevation gradient favors *Artemisia* spp. and Cyperaceae for NAP and *Salix* spp. and *Larix sibirica* for AP (Fig.5.D).

### 4.1.3    MAT and WAPLS Results

To reconstruct bioclimatic parameters from pollen data, MAT and WAPLS methods were applied on the four scales, modern pollen datasets and the ten climate parameters (Table 1). All these methods can be run with $n \in [1; 10]$ parameters: the number of analogues for MAT and the number of components for WAPLS. The best transfer functions among all of them were selected by the following approach: in a first step, for each climate parameter the methods fitting with the higher $R^2$ and the lower RMSE were selected. Then, in case the highest $R^2$ and the lowest RMSE were not applied for the same number of analogues

or components, we chose the method presenting the lower number of parameters. Despite the small number of parameters relative to the number of observations, the method fits well (Arnold, 2010, table 1). MAT transfer function gives better $R^2$ in bigger DB than in smaller ones. Fitting increases with the diversity and the size of DB, since MAT is looking for the closest value between climate and pollen abundance. By contrast, WAPLS fits better on the local scale and especially with a smaller number of sites. In this case, the pull of data is largest and the variance is largest (Ter Braak and Juggins, 1993). WAPLS also

leads to better value of RMSE than $R^2$, in contrast to MAT. For temperature, pollen fits better with $T_{spr}$ or MTWA in Mongolia. Temperatures of the warmest months indeed control both vegetation extension and pollen production (Ge et al., 2017; Li et al., 2011) and especially in very cold areas such as Mongolia. For precipitation, the significant season is the one associated with the Summer Monsoon System in Mongolia (Wesche et al., 2016). Almost all the Mongolian precipitation falls during the spring and the summer (Wang et al., 2010), and the amount of precipitation controls, among other parameters, the openness

of the landscape in Mongolia (Klinge and Sauer, 2019). To simplify the confrontation of the diverse models, the MAAT and MAP were isolated from the rest of the climate parameters. Even if these two climate parameters are not the best fitting pollen methods, they are the easiest to interpret and are comparable with the GDGT regression models commonly based on MAAT and MAP.

## 4.2    GDGT – Climate Calibration

### 4.2.1    GDGT Variance in the dataset

Iso–GDGTs are dominated by $GDGT_0$ and crenarcheol (X percent in relative abundances, respectively, in Fig.4.A). Since the majority of these molecules are thought to be produced in the lake water column (Schouten et al., 2012), the variations of fractional abundance in the soils and moss samples are very discrete and poorly linked to climate parameters. Br–GDGT





concentrations differ depending on the sample type:

$$[\mathrm{br--GDGT_{tot}}]_{\mathrm{sed}} = 88 \pm 18 \mathrm{ng.g_{sed}^{-1}} \quad (2)$$

$$[\mathrm{br--GDGT_{tot}}]_{\mathrm{moss}} = 82 \pm 76 \mathrm{ng.g_{moss}^{-1}}$$

$$[\mathrm{br--GDGT_{tot}}]_{\mathrm{soil}} = 24 \pm 30 \mathrm{ng.g_{soil}^{-1}}$$

br–GDGT fractional abundances are consistent with each type of sample: the major compounds are the $I_a$, $II'_a$, $II_a$ and $III_a$

(Fig. 4B). These compounds explain dominantly the total variance (Fig. 5A). Particularly, the $PC_1$ representing 22.77 % of the total variance identifies two clusters: the 5-methyl associated with the taiga samples on one side and the 6, 7-methyl associated with the steppe and desert samples on the other side

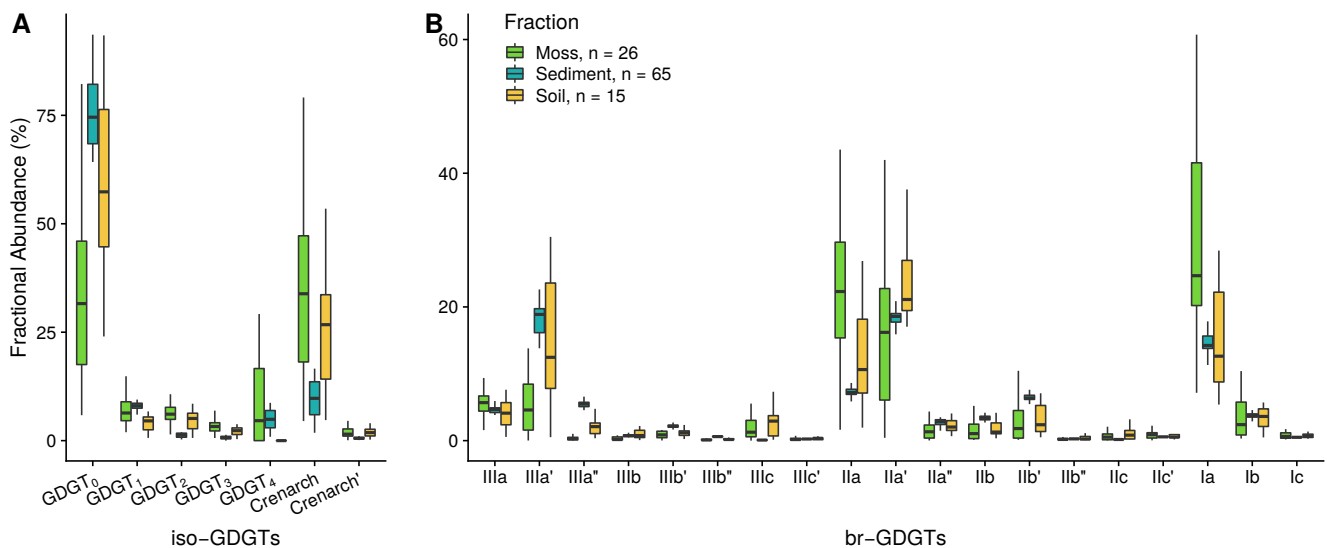

**Figure 4.** Fractional abundances of (**A**) isoprenoid-GDGTs and (**B**) branched-GDGTs for moss polsters (green), soil surface samples (orange) and lacustrine core top sediments (blue). The punctuation marks ' and " refer to 6 and 7 methyl, respectively.

The sediment samples from Mongolian lakes are more homogeneous than the surface samples, especially when compared

with the moss polsters that present a wide variability (Fig. 4B). Generally, soil samples are more relevant analogues to sediments than moss polsters (Fig. 4B). This variability shows an influence of the sample type on br–GDGT responses. Sample type also bears climate information, since soil and moss polsters originate from steppe to desert environments and forest/alpine meadows, respectively.

### 4.2.2 Climate Influence on br–GDGT Indexes

The influence of the bio-climate parameters on the br–GDGT matrix variance is connected with MAP (Fig. 5.B: $RDA_1 = 10.01\%$). The negative values show higher precipitations and uncycled 5-Me GDGTs, such as $I_a$, $II_a$ and $III_a$. While the lower MAP





**Figure 5.** Multivariate statistics for the proxies clustered by ecosystems. **A**: Principal Components Analysis (PCA) and (**B**) Redundancy Analysis (RDA) for br–GDGT fractional abundances; **C**: PCA and (**D**) RDA for pollen fractions. The variance percentage explained is displayed on the axis label; the size of the dataset (n) and the RDA linear regression ($R^2$) are inserted in each plot area.





match with 6 or 7-Me GDGTs, such as $\mathrm{III}''_a$, $\mathrm{II}''_a$, $\mathrm{II}'_a$. The $\mathrm{RDA}_2$ is slightly more connected to MAAT and elevation, also clustering the methyled and cyclized GDGTs to the higher MAAT. The correlation between chemical structure and climate parameters (Weijers et al., 2004; Huguet et al., 2013) was not strong. All the MBT, MBT', $\mathrm{MBT}'_{5Me}$ and CBT, CBT' , $\mathrm{CBT}_{5Me}$

relations with climate parameters were tested and showed very low correlation with $R^2 \in [0.1; 0.35]$ (Supplementary Figure S3).

### 4.2.3 Multi-regression Models

The Stepwise Selection Model for the climate – br–GDGT modelling was applied only on the 5 and 6 Methyls, because 7-methyled br–GDGTs show weak significance in the variance explanation (PCA, Fig. 5A). The $\mathrm{N_{SSM}}$ different combinations

of the 15 br–GDGT molecules result in $\mathrm{N_{SSM}} = 2^{15} = 32768$ models possible for each climate parameter. The better fitting equations (with low RMSE and AIC and high R-squared) were selected for each number of parameters (number of br–GDGT issued in the linear regression) for both MAAT and MAP. Within the 2 series of 15 models, only 9 were selected for discussion (Table 2). The models with the best statistical results were comprised of between 5 to 12 parameters and present a $R^2 \in [0.60; 0.76]$, a RMSE around 1.1 °C or 70 $mm.yr^{-1}$ and a $\mathrm{AIC_{MAAT}} \in [152.6; 166.2]$ or $\mathrm{AIC_{MAP}} \in [152.6; 166.2]$. The

earlier a parameter is used in the MR models, the more influence it has. For both $\mathrm{MAAT_{mr}}$ and $\mathrm{MAP_{mr}}$ models, $\mathrm{III}_a$, $\mathrm{III}'_a$, $\mathrm{III}_b$ and $\mathrm{III}'_b$ are the most relevant br–GDGT fractions for the climate reconstruction (Table 2) which is consistent with the PCA and RDA observations displayed in Fig. 5A and B. The $\Delta T$ values closest to 0 reveal the best fitting model on each point (Fig. 7, panel 1). Then, the box-plot (Fig. 7, panel 2) summarises the best fitting model at the regional scale.

## 5 Discussion

### 5.1 Over-Parameterization and Selection of the Best Methods

Among the possible methods, statistical values help to select the most reliable ones for palaeoclimate reconstruction. However, the correlation ($R^2$) and errors (RMSE) are not enough to discriminate between them to identify the most suitable ones for palaeoclimate modelling, especially for the multi-parameter methods, such as br–GDGT multi-regression models and pollen transfer functions. Indeed, the more input parameters in the method, the more accurate it is (Tables 1, 2 and Fig. 6A and 6B). All

the regression models improve with parameter additions, and especially the less fitting methods improve exponentially (lower limit of the $R^2$ area, Fig. 6B). The best R-squared-models for each parameter number (Fig. 6A) correspond to the upper limit of the $R^2$ area (Fig. 6B). The $R^2$ trend in function of the parameter number follows a logistic regression both for $\mathrm{MAAT_{mr}}$ and $\mathrm{MAP_{mr}}$ models. However, and especially for $\mathrm{MAAT_{mr}}$ regression models, this logistic curve becomes asymptotic early, similar to the RMSE decrease. This model over-parameterization has proven to produce artefacts in ecological modelling

(Arnold, 2010; Symonds and Moussalli, 2011). The issue is thus to identify the threshold in the parameter numbers selected. We used Akaike's Information Criterion (AIC) to determine the better model without over-parameterization for br–GDGT regression models: the lower the AIC, the better the model (Table 2). The trend of AIC versus the parameter number is however

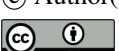



more complex (Fig. 6C). For $MAAT_{mr}$, the regression model becomes more accurate from one to five parameters quite rapidly, but then slowly decreases. The AIC curve takes an asymmetrical hollow shape around five parameters with a steeper slope on the left side (Fig. 6A). The AIC values for $MAAT_{mr6}$ and $MAAT_{mr7}$ are almost identical (Fig. 6A) . The $MAP_{mr6,7,8}$ have almost equivalent AIC values, while the AIC curve shapes differ for the other $MAP_{mr}$ models (asymmetrical hollow shape around five with a steeper slope on the left side, Fig. 6A). To sum-up, the most universal models are $MAAT_{mr5}$ and $MAP_{mr7}$ but the closed models are also valuable in some local contexts. We need to determine the cross-values of these models to select the appropriate ones for the Siberian-Mongolian context.

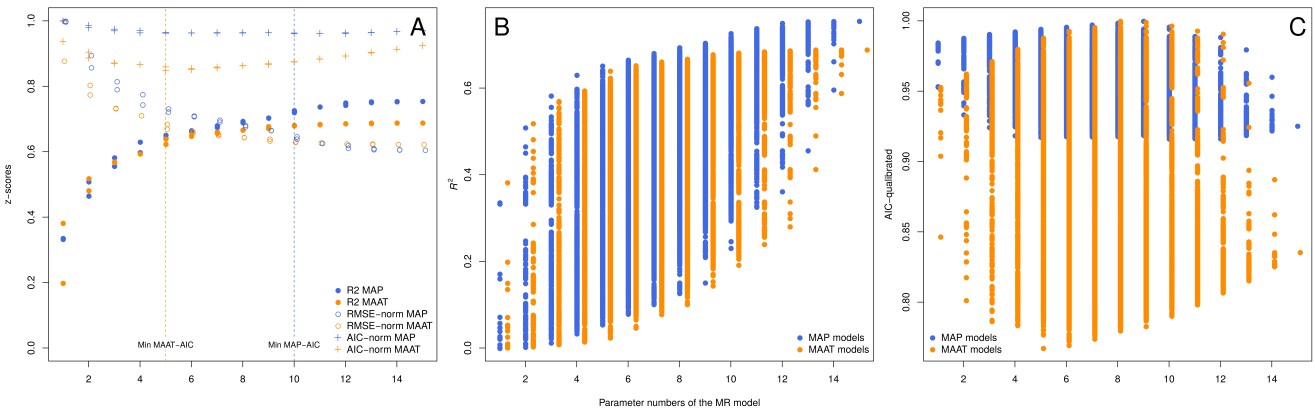

**Figure 6.** Statistical values plotted against the number of parameters of the different MR-GDGT models: the $R^2$, the RMSE normalized on the highest RMSE value and the AIC also normalized. **A**: selection of the two best MR models for each number of parameters; **B**: combination of the $R^2$ (**B**) and the AIC (**C**) values for of all the MR models. The blue dots are for the MAP-models, orange dots for the MAAT one.

## 5.2   Assessment of the Calibration Feedback

The cross-values of the nine best $MAAT_{mr}$ regression models (Fig. 7A1 and 7A2) and the best $MAP_{mr}$ regression models (Fig. 7B1 and 7B2) were tested. The MAAT reconstructions provide different responses to the three main bio-climate areas (parcel A1): if they properly estimate temperatures in the Siberian Baikal basin, they overestimate and underestimate them for the center of the northern Mongolian mountains and the Gobi desert, respectively. For precipitation (parcel B1), all the $MAP_{mr}$, including the ones based on the local to regional databases, also misrepresent the extreme values: precipitation values are too high and too low for the Gobi desert and the Baikal basin, respectively. To conclude, the wider the dataset extension, the more alleviated the extreme values.

Both on $MAAT_{mr}$ and $MAP_{mr}$ models, the 95 % interval shrinks with parameter addition, but the mean values do not necessarily get closer to the climate parameter measured value (the dashed line in Fig. *7.A2 and B2*). Therefore, if the tests on the AIC point toward the $MAAT_{mr4}$ and the $MAP_{mr7}$ regression models, the back-cross plots suggest the $MAAT_{mr3}$ and the





$\text{MAP}_{\text{mr6}}$ regression models provide the best estimates for climate reconstruction in lacustrine archives ($\Delta\text{MAP} = 0$ and best fitting temperature for the mean value of all samples, Fig. 7.B2 and Fig. 7.B1).

### 5.3 Global vs. Local Calibration

Whatever proxy is used, when reconstructing temperatures and precipitation from past archives in a given location, there is the issue of basing reconstructions on calibrations based on local or global datasets (among others, Tian et al., 2014; Cao et al., 2014; Ghosh et al., 2017, , Tierney et al., 2019 in press). We tested both approaches on our datasets. The global br–GDGT - climate calibration artificially reaches higher R-squared than local ones due to the larger range of values of the climate parameters involved ($MAAT \in [-5; 30]$ in Naafs et al. (2017) soil world dataset against $MAAT \in [0; 5]$ in this case study,

inducing a lower signal/noise ratio). Despite the relatively lower R-squared value of 0.62 scored by the $\text{MAAT}_{\text{mr5}}$ inferred model compared with (Pearson et al., 2011; De Jonge et al., 2014a; Naafs et al., 2017), the boxplots for the all $\text{MAAT}_{\text{mr}}$ calculated from the NSMDB are mostly centred on the mean MAAT value with the shortest variance spreading for all the sites (Fig. 7C1 and 7C2). These local calibrations fit the best with the $\text{MAAT}_{\text{Ding}}$ from Ding et al. (2015) which is also a local calibration made on the Tibet-Qinghai plateau database. The global databases made on worldwide sites (De Jonge et al.,

2014b, a; Naafs et al., 2017) provide $\text{MAAT}_{\text{model}} > \text{MAAT}_{\text{real}}$ and large standard deviation (SD). The extreme MAAT values are attenuated with reconstructed temperatures higher by up to +5 to +10 °C in the Siberian basin and the Mongolian plateau, and lower by up to to -3 to - 5 °C in the Gobi desert. Finally, the local calibrations performed on subtropical to tropical Chinese transects (Yang et al., 2014; Thomas et al., 2017) have smaller SD but largely overestimate MAAT values due to the warmer conditions of the initial database sites.

Similarly, for pollen transfer functions, the geographic range of the surface samples on which the calibration relies is a relevant parameter to take into account for the trustworthiness of the paleoclimate reconstructions. The choice of the maximum value of this geographic range has been discussed previously for vegetation modelling, for example, the Relevant Source Area of Pollen (RSAP, Hellman et al., 2009a, b; Bunting and Hjelle, 2010; Prentice, 1985). For MAT and WAPLS regression models, the same issue holds true. The responses of the eight over-represented *taxa* to climate parameters are different in the three

geographic ranges. The linear tendency allows for checking the main trends between taxa distribution and climate parameters, despite the weak linear regressions ($p-value > 0.005$ and $R^2 < 0.4$, in Fig. 8) For the majority of these *taxa*, the trend is the same, independent of the database size (*Larix* spp. and Cyperaceae percentages increasing with weaker MAAT, or Amaranthaceae and *Pinus sylvestris* percentages increasing with higher MAAT). However, due to the shift between pollen types and their associated vegetation (i.e. poaceae-pollen signal similar for a wide diversity of Poaceae communities with very contrasted

ecophysiological features), trends are controlled in some peculiar cases by the geographical clipping of the DB. Thus, Poaceae have a positive response to MAP on the global scale but not inside the Mongolian area. The human influence on pollen rain is also dependent on the biogeographical context, thus, *Artemisia* spp. is not considered as much human influenced in the Asian steppe environment (Liu et al., 2006) than in the European one (Brun, 2011).





**Figure 7.** Validation of br–GDGT-climate models on the study sites: reconstructed values for literature MAAT calibrations (**A**), for MR-MAAT -this study- (**B**) and for MR-MAP -this study- (**C**). Models are tested for each sample of the NMDB (**1**) and their box-plot statistical values (**2**) are provided. Sites are clustered in four groups from left to right: core-top lacustrine samples of the MR model dataset; moss polsters from Mongolian steppe-forest plateau; Gobi steppe-desert soil samples and moss polsters from Baikal basin. Values are plotted in anomaly: $\Delta = MAAT_{real} - MAAT_{model}$.



**Figure 8.** Relationships between the eight major pollen *taxa* (%TP) and MAAT (°C, upper part of the facet plot) and MAP ($mm.yr^{-1}$, lower part). The black line is the linear fitting for all samples (COSTDB), the orange for all the samples from Mongolia (MDB) and the blue only for the NSMDB samples presented in this article.



Concerning modeling transfer functions, WAPLS performs better for the local database than for the MDB and COST database (Table 1). On these subsets, the WAPLS RMSE and R-square values are even higher than for the MAT transfer function. The major difficulty resides in the reconstructions of precipitation. Even if the RMSE and $R^2$ values are higher for all models of MAP than MAAT, the influence of precipitation on vegetation cover is not well understood. In Mongolia it is clear that the precipitation controls the treeline in mountainous areas (Klinge and Sauer, 2019) and the global openness in the steppe

- forest ecotone (Wesche et al., 2016) as well as human land-use (Tian et al., 2014), but the risk of autocorrelation between MAAT and MAP signals is important, even if the RMSE and $R^2$ values are higher for MAP regression models than for MAAT ones (Telford and Birks, 2009; Cao et al., 2014).

### 5.4   Issues in Modelling Mongolian Extreme Bioclimate

Firstly, the commonly used br–GDGT indexes (MBT and CBT) are not relevant for arid areas with $\mathrm{MAP} < 500\mathrm{mm.yr}^{-1}$

because of the relation between low soil water content and soil br–GDGT preservation and conservation interferes in the br–GDGT / climate parameters (Dang et al., 2016). Moreover, the main issue in climate modelling in Mongolia is the narrow relationship between MAAT and MAP. Because of the climatic gradient from dry deserts in the southern latitudes to wet taiga forests in the northern ones, MAAT and MAP maps are strongly anti-correlated (Fig. 1 B and C). This correlation could also be a bias resulting from the interpolation method of the *WorldClim2* database. In fact, there are very few weather stations

(Fig. 1.A, Fick and Hijmans, 2017) on the large Mongolian plateau area and a great diversity of mountain ranges interrupting them. Moreover, the relevance of the interpolation models suffers from the transition threshold made by Mongolia between the EASM and the Siberian Westerlies (Fig.2, An et al., 2008).

However, both GDGT and pollen models show that the precipitation calibrations are more reliable than temperature ones

(Tables 2, 1, Figures 3, 6 and 7), reflecting that the Siberian-Mongolian system seems to be controlled by precipitation. This dominance of precipitation could be due to seasonality. Even if the br–GDGT production is considered to be mainly linked to MAAT (Weijers et al., 2007a, b; Peterse et al., 2012), the high pressure Mongolian climate system (Zheng et al., 2004; An et al., 2008) favors a strong seasonal contrast: almost all the precipitation and the positive temperature values happen during the summer (Wesche et al., 2016). Consequently, for the NMSDB pollen models (Table 1). The seasonal parameters such

as MTWA, $\mathrm{T_{sum}}$ and $\mathrm{P_{sum}}$ better describe the GDGT variability than MAAT and MAP. It is the opposite on EAPDB and COSTDB models. The Mongolian permafrost persists half the year in the northern part of the country (Sharkhuu, 2003) and acts on vegetation cover and pollen production (Klinge et al., 2018). Furthermore, the effects of frozen soils on soil bacterial communities and GDGT production are thought to be important (Kusch et al., 2019). This system leads to a quasi equivalence between MAP and $\mathrm{P_{Sum}}$ while MAAT is torn apart by the large $\mathrm{T_{Sum} - T_{win}}$ contrast. The MAP appears then to be the most

reliable climate parameter for Siberian-Mongolian climate studies under the threshold of about 5 °C.

To reduce the signal/noise ratio, a wider diversity of sample sites should be added as initial inputs in the models. This raises the question of the availability of reliable samples in desert areas. The soil samples in the steppe to desert biomes are often





very dry and these over-oxic soil conditions are the worst for both pollen preservation (Li et al., 2005; Xu et al., 2009) and
GDGT production (Dang et al., 2016). br–GDGT concentrations in moss polsters and sediments are thus higher than in soils
in our database.

The soil of the Gobi desert also has a high salinity level which is also a parameter of control on br–GDGT fractional
abundance (Zang et al., 2018). Even if the impact on sporopollenin is not well understood, the salinity also affects pollen
conservation in soils (Reddy and Goss, 1971; Gul and Ahmad, 2006). This taphonomic bias (also climatically induced) could
explain part of the histogram variance of Fig. 4 related to the sample type as well as the shift of the soil–cluster from the
regression line in the cross-value plot of br–GDGT MBT'/CBT models in Supplementary Figure (S3)

Finally, the saturation effect of the proxies when they reach the limits of their range of appliance is also taken into account.
Since both pollen and br–GDGT signals are analysed in fractional abundance (i.e. % of the total count of concentration), these
proxies evolve in a $[0; 1]$ space. The saturation effect appears when extreme climatic conditions are reached (Naafs et al., 2017).
For instance, in a tropical context, temperature values are too high to be linearly linked to fractional abundances (Pérez-Angel
et al., 2019). Considering pollen–climate relationships, the inferior limit of pollen percentage is critical: for the majority of
pollen types, whenever the MAAT or MAP reaches a very high or low threshold, the pollen fraction approaches zero (Fig. 8).
These limit areas need to be investigated closely, which legitimises the local calibration methods.

## 6    Conclusions

The palaeoenvironmental and palaeoclimatic signals present several uncertainties which can misguide the interpretation of
past variations. This study shows how both a multi-proxy approach and accurate calibration are important in combating these
biases. We propose a new calibration for Mean Annual Precipitation (MAP) and Mean Annual Air Temperature (MAAT) from
br–GDGTs as well as a new pollen surface database available for transfer functions. The correlations between pollen rain and
climate on one hand and br–GDGT soil production and climate on the other are visible but are still mitigated by the complex
climate system of arid central Asia and the diversity of soils and ecosystems. Precisely, each of our proxies seems to be more
narrowly linked to precipitation (MAP) than temperature (MAAT) counter to the majority of calibrations in the literature.
The nature of the samples considered (soil, moss polster and sediment top-core) also greatly affected these correlations. The
calibration work in the extreme bio-climates of the Siberian basin and Mongolian plateau is difficult because of the low
range of climate values, despite the climate diversity ranging from cold and slightly wet (north) to the arid and warm (south).
The MAAT and MAP values do not remarkably spread in the vectorial space, which makes harder to distinguish the linear
correlation against variance noise. Moreover, this range of values is close to the lower saturation limit of the proxies, which
makes the accurate local calibration tricky but necessary. The local calibrations also suffer from the reduced size and small
geographic extent of the dataset. The vegetation cover, extending from a high cover taiga forest to nude soil desert cover, also
buffers the climate signal and the GDGT / pollen response. The correlations between climate parameters and GDGT / pollen
proportion are therefore lower than they could be at the global scale. Nonetheless, and despite the lower correlation of the local calibration, these local approaches appear to be more accurate to fit with the actual climate parameters than the global ones: both for pollen function transfer and br–GDGT multiple regression models. These positive model results have to be considered

in front of over-parameterization limits. Too many parameters in MR–GDGT models or in pollen MAT or WAPLS transfer function can add artificially to the linear relation between climate and proxies and lead to misinterpretation of palaeoclimate records. Akaike's information criterion associated with RMSE and $R^2$ values is a fair way to select the best climate model. We encourage the wider application of this local multi-proxy calibration for a more accurate constraint of these central Asian climatic systems, a crucial improvement to properly model the fluctuations of the Monsoon Line since the Optimum Holocene.

**Appendix A:  Appendix tables**

The new calibration of climate reconstruction for Mongolia and Siberia presented in this study is based up on the New Mongolia-Siberia Data Base (NMSDB). Location, ecosystems as well as sample type are provided in Table (S1). Published br–GDGT/climate calibration equations are tested on the (NMSDB) (S2).

**Appendix B:  Appendix figure**

Cross-plots allow to evaluate the efficiency of these methods (fig. S3).

*Author contributions.*  LD conducted the analytical work, LD, SJ, OP, GM designed the study. All the authors contributed to the scientific reflection as well as to the preparation of the manuscript.

*Competing interests.*  The authors declare having no competing interests

*Acknowledgements.*  We want to thank all the direct and indirect contributors to the global surface pollen dataset as well as the Laboratory of
Ecological and Evolutionary Synthesis of the National University of Mongolia for its support during the field trip. We also express gratitude to Laure Paradis for her GIS advice, Marc Dugerdil for the help with Python fixing, Jérôme Magail and the Monaco–Mongolia joint mission for their technical and financial support in providing top cores and sediment samples from Arkhangai, and Salomé Ansanay-Alex for her spectrometer expertise. We are grateful to the ISEM team DECG for financial support. For the analytical work completed at LGLTPE-ENS de Lyon, this research was funded by Institut Universitaire de France funds to GM..
improve this manuscript. This is an ISEM publication n°: XXX



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





**Table 1.** Statistical results of the MAT and WAPLS methods applied to four surface pollen datasets and ten climate parameters[a].

| Model | Database | Climate parameter | k best R$^2$[b] | k best RMSE[b] | k selected[c] | R$^2$ selected[c] | RMSE selected[c] |
|---|---|---|---|---|---|---|---|
| **WAPLS** | **NMSDB** | MAAT | 2 | 2 | 2 | 0.66 | 1.17 |
| | | MTWAQ | 2 | 2 | 2 | 0.62 | 1.68 |
| | | Tspr | 2 | 2 | 2 | 0.70 | 1.27 |
| | | MAP | 2 | **1** | 1 | 0.79 | 61.67 |
| | | Pspr | 2 | **1** | 1 | 0.66 | 11.5 |
| | | Psum | 2 | 2 | 2 | 0.80 | 34.8 |
| | **MDB** | MAAT | 2 | **1** | 1 | 0.35 | 1.91 |
| | | MTWA | 2 | **1** | 1 | 0.24 | 2.14 |
| | | MTCO | 2 | **1** | 1 | 0.26 | 2.75 |
| | | MAP | 3 | **1** | 1 | 0.23 | 95.11 |
| | | Psum | 1 | 1 | 1 | 0.47 | 46.86 |
| | **COSTDB** | MAAT | 2 | 2 | 2 | 0.54 | 4.10 |
| | | MTWA | 3 | **2** | 2 | 0.48 | 3.55 |
| | | MTCO | 2 | 2 | 2 | 0.56 | 6.34 |
| | | MAP | 4 | **2** | 2 | 0.55 | 223.67 |
| | | Psum | 3 | **2** | 2 | 0.34 | 70.8 |
| | **EAPDB** | MAAT | 3 | 3 | 3 | 0.72 | 4.08 |
| | | MTWA | 3 | 3 | 3 | 0.55 | 3.31 |
| | | MTCO | 3 | 3 | 3 | 0.72 | 6.49 |
| | | MAP | 3 | 3 | 3 | 0.43 | 239.6 |
| | | Psum | 2 | 2 | 2 | 0.52 | 62.33 |
| **MAT** | **NMSDB** | MAAT | **2** | 10 | 2 | 0.59 | 1.49 |
| | | MTWAQ | **2** | 9 | 2 | 0.68 | 1.80 |
| | | Tspr | **2** | 10 | 2 | 0.62 | 1.66 |
| | | MAP | **2** | 6 | 2 | 0.88 | 55.52 |
| | | Pspr | **2** | 4 | 2 | 0.89 | 8.26 |
| | | Psum | **2** | 9 | 2 | 0.82 | 37.82 |
| | **MDB** | MAAT | **5** | 8 | 5 | 0.61 | 1.64 |
| | | MTWA | **6** | 9 | 6 | 0.53 | 1.84 |
| | | MTCO | **5** | 7 | 5 | 0.66 | 2.05 |
| | | MAP | **9** | 10 | 9 | 0.38 | 88.72 |
| | | Psum | **8** | 10 | 8 | 0.54 | 46.3 |
| | **COSTDB** | MAAT | **8** | 9 | 8 | 0.73 | 3.28 |
| | | MTWA | **8** | 10 | 8 | 0.67 | 2.96 |
| | | MTCO | **6** | 8 | 6 | 0.78 | 4.78 |
| | | MAP | **6** | 8 | 6 | 0.78 | 166.17 |
| | | Psum | **6** | 9 | 6 | 0.66 | 53.55 |
| | **EAPDB** | MAAT | **5** | 8 | 5 | 0.88 | 2.90 |
| | | MTWA | **5** | 9 | 5 | 0.79 | 2.50 |
| | | MTCO | **4** | 8 | 4 | 0.89 | 4.46 |
| | | MAP | **4** | 10 | 4 | 0.74 | 181.21 |
| | | Psum | **4** | 8 | 4 | 0.80 | 44.66 |

[a] Only the five better fitting regression models for each climate parameter are shown .

[b] Corresponding to the number of parameters used in the model inferring the best $R^2$ and $RMSE$.

[c] Number of parameters, $R^2$ and $RMSE$ of the finally selected model.



**Table 2.** Statistical values and equations of the best br–GDGT $MAAT_{mr}$ and $MAP_{mr}$ models.

| Model | k | $R^2$ | RMSE | AIC | Formula |
|---|---|---|---|---|---|
| $MAAT_{mr1}$ | 1 | 0.38 | 1.5 | 168.4 | $MAAT_{mr1} = -0.5 \times 1 + 12.9 \times [IIIa']$ |
| $MAAT_{mr2}$ | 2 | 0.52 | 1.4 | 162.6 | $MAAT_{mr2} = -0.7 \times 1 + 13.4 \times [IIIa'] + 11.8 \times [IIIb]$ |
| $MAAT_{mr3}$ | 3 | 0.57 | 1.3 | 156.4 | $MAAT_{mr3} = 0.6 \times 1 - 25.1 \times [IIIa] + 12.3$ $\times [IIIa'] + 7.2 \times [Ib]$ |
| $MAAT_{mr4}$ | 5 | 0.62 | 1.2 | **152.6** | $MAAT_{mr4} = 4.5 \times 1 - 36.8 \times [IIIa]$ $+ 7.3 \times [IIIa'] - 37.2 \times [IIIc] - 24 \times [IIb] - 5.2 \times [Ia]$ |
| $MAAT_{mr5}$ | 7 | 0.66 | **1.1** | 153.9 | $MAAT_{mr5} = 4.8 \times 1 - 38.5 \times [IIIa] + 7.9 \times [IIIa']$ $- 27.3 \times [IIIc] - 3.3 \times [IIa'] - 26.3 \times [IIb] + 8.5 \times [IIb']$ $- 5.6 \times [Ia]$ |
| $MAAT_{mr6}$ | 9 | 0.67 | **1.1** | 155.7 | $MAAT_{mr6} = 12.3 \times 1 - 52.1 \times [IIIa] - 16.9 \times [IIIb]$ $- 25.9 \times [IIIb'] - 41.1 \times [IIIc] - 6 \times [IIa] - 10.4 \times [IIa']$ $- 38.5 \times [IIb] - 13.3 \times [Ia] - 32.8 \times [Ic]$ |
| $MAAT_{mr7}$ | 10 | 0.68 | **1.1** | 157.2 | $MAAT_{mr7} = 12.3 \times 1 - 52.6 \times [IIIa] - 16.8 \times [IIIb]$ $- 25.3 \times [IIIb'] - 35.7 \times [IIIc] - 6 \times [IIa] - 10.5 \times [IIa']$ $- 37.8 \times [IIb] - 15.4 \times [IIc] - 13.2 \times [Ia] - 31.4 \times [Ic]$ |
| $MAAT_{mr8}$ | 12 | **0.69** | 1.1 | 160.5 | $MAAT_{mr8} = 12.5 \times 1 - 54.9 \times [IIIa] - 23.6 \times [IIIb]$ $- 26.8 \times [IIIb'] - 35.1 \times [IIIc] - 23.4 \times [IIIc'] - 5.9 \times [IIa]$ $- 10.4 \times [IIa'] - 40.6 \times [IIb] - 16 \times [IIc] - 13.5 \times [Ia]$ $+ 5.6 \times [Ib] - 35.8 \times [Ic]$ |
| $MAAT_{mr9}$ | 15 | **0.69** | 1.1 | 166.2 | $MAAT_{mr9} = 10.4 \times 1 - 49.5 \times [IIIa] + 3.3 \times [IIIa']$ $- 21.5 \times [IIIb] - 26.2 \times [IIIb'] - 30.9 \times [IIIc] - 23.3 \times [IIIc']$ $- 4.4 \times [IIa] - 8.9 \times [IIa'] - 38.1 \times [IIb] + 0.3 \times [IIb']$ $- 13.2 \times [IIc] + 0 \times [IIc'] - 11.1 \times [Ia] + 7.7 \times [Ib] - 31.7 \times [Ic]$ |
| $MAP_{mr1}$ | 1 | 0.34 | 112 | 546.1 | $MAP_{mr1} = 179.3 \times 1 + 509.1 \times [Ia]$ |
| $MAP_{mr2}$ | 2 | 0.51 | 96.1 | 534.6 | $MAP_{mr2} = 59.6 \times 1 + 2289.9 \times [IIb'] + 710.8 \times [Ia]$ |
| $MAP_{mr3}$ | 3 | 0.58 | 91.3 | 532.1 | $MAP_{mr3} = 94.1 \times 1 + 2319.7 \times [IIb'] + 702.2 \times [Ia]$ $- 3752.5 \times [Ic]$ |
| $MAP_{mr4}$ | 5 | 0.65 | 82 | 526.7 | $MAP_{mr4} = 245.1 \times 1 - 666.7 \times [IIIa'] + 2431.9 \times [IIb']$ $+ 396.2 \times [Ia] - 276 \times [Ib] - 3780 \times [Ic]$ |
| $MAP_{mr5}$ | 8 | 0.69 | 75.9 | 525.8 | $MAP_{mr5} = -103.8 \times 1 + 1553.4 \times [IIIa] + 536 \times [IIIb]$ $- 3145.6 \times [IIIb'] + 2480.3 \times [IIIc] + 1290.1 \times [IIb]$ $+ 2459.3 \times [IIb'] + 927.9 \times [Ia] - 2955.3 \times [Ic]$ |
| $MAP_{mr6}$ | 10 | 0.73 | 72.5 | 525.8 | $MAP_{mr6} = -511.3 \times 1 + 1205.9 \times [IIIa] + 1387.2 \times [IIIb]$ $+ 738.8 \times [IIa] + 969.8 \times [IIa'] + 1957.1 \times [IIb]$ $+ 3006.3 \times [IIb'] + 2406.4 \times [IIc] + 2003.1 \times [IIc']$ $+ 1081.7 \times [Ia] - 2406.3 \times [Ic]$ |
| $MAP_{mr7}$ | 12 | 0.75 | 68.5 | **524.8** | $MAP_{mr7} = -502.6 \times 1 + 1359.5 \times [IIIa] + 2462.5 \times [IIIb]$ $- 2178.3 \times [IIIb'] + 657.7 \times [IIa] + 986.8 \times [IIa']$ $+ 2440.5 \times [IIb] + 3423.5 \times [IIb'] + 2831.2 \times [IIc]$ $+ 1967.2 \times [IIc'] + 1150.2 \times [Ia] - 955.6 \times [Ib] - 2103.4 \times [Ic]$ |
| $MAP_{mr8}$ | 15 | **0.76** | **67.9** | 530 | $MAP_{mr8} = -619.5 \times 1 + 1725.6 \times [IIIa] + 161.7 \times [IIIa']$ $+ 2603.5 \times [IIIb] - 2069.7 \times [IIIb'] + 380.3 \times [IIIc]$ $+ 2226.9 \times [IIIc'] + 730.5 \times [IIa] + 1028.2 \times [IIa']$ $+ 2569 \times [IIb] + 3424.2 \times [IIb'] + 2734.6 \times [IIc] + 1830.5 \times [IIc']$ $+ 1289.3 \times [Ia] - 854.1 \times [Ib] - 1745.6 \times [Ic]$ |



**Table A1.** Sample sites included in the New Mongolia-Siberia DataBase (NMSDB).

| Label | Lat. | Long. | Elev. | Pollen | GDGT | Type | Biomes |
|---|---|---|---|---|---|---|---|
| MMNT1M01 | 48.3983 | 106.8594 | 1137 | x | x | Moss | Steppe-forest |
| MMNT1M02 | 48.4014 | 106.8613 | 1161 | x | x | Moss | Steppe-forest |
| MMNT2M01 | 48.4472 | 107.0542 | 1438 | x | x | Moss | Steppe-forest |
| MMNT2M02 | 48.4460 | 107.0551 | 1333 | x | x | Moss | Steppe-forest |
| MMNT2M03 | 48.4449 | 107.0564 | 1265 | x | x | Moss | Steppe-forest |
| MMNT2M04 | 48.4441 | 107.0580 | 1266 | x | x | Moss | Light taiga |
| MMNT2M05 | 48.4425 | 107.0593 | 1262 | x | | Moss | Light taiga |
| MMNT2M05' | 48.4444 | 107.0634 | 1273 | x | x | Moss | Light taiga |
| MMNT2M06 | 48.4412 | 107.0629 | 1328 | x | x | Moss | Light taiga |
| MMNT2M07 | 48.4381 | 107.0660 | 1475 | x | x | Moss | Steppe-forest |
| MMNT3S01 | 47.2993 | 103.6092 | 1323 | x | x | Soil | Steppe |
| MMNT3S02 | 47.2000 | 102.8438 | 1457 | x | x | Soil | Steppe |
| MMNT3M03 | 46.8239 | 102.2307 | 1669 | x | x | Moss | Alpine meadow |
| MMNT3M04 | 46.7932 | 102.0868 | 1734 | x | x | Moss | Alpine meadow |
| MMNT3S05 | 46.7800 | 101.9510 | 1830 | x | x | Soil | Alpine meadow |
| MMNT4S01 | 45.6645 | 101.6054 | 1750 | x | x | Soil | Steppe-desert |
| MMNT4S02 | 45.1759 | 101.4288 | 1238 | x | x | Soil | Desert |
| MMNT4S03 | 45.1724 | 101.4517 | 1233 | x | x | Soil | Desert |
| MMNT4S04 | 45.1702 | 101.4806 | 1230 | x | x | Soil | Desert |
| MMNT4S05 | 45.1618 | 101.4927 | 1228 | x | x | Soil | Desert |
| MMNT4S06 | 45.1467 | 101.5083 | 1230 | x | x | Soil | Desert |
| MMNT4S07 | 45.1402 | 101.5087 | 1232 | x | x | Soil | Desert |
| MMNT4S08 | 44.6746 | 102.1844 | 1508 | x | x | Soil | Steppe-desert |
| MMNT4M09 | 44.4509 | 102.3459 | 1847 | x | x | Moss | Steppe-desert |
| MMNT4M10 | 44.3952 | 102.4511 | 1677 | x | x | Moss | Steppe-desert |
| MMNT4S11 | 44.1685 | 102.6031 | 1273 | x | x | Soil | Desert |
| MMNT4S12 | 43.9494 | 102.7411 | 1574 | x | x | Soil | Steppe-desert |
| MMNT4S13 | 43.8636 | 102.7479 | 1802 | x | | Soil | Steppe-desert |
| MMNT4S14 | 43.7650 | 102.8018 | 1982 | x | x | Soil | Steppe-desert |





**Table A2.** Sample sites included in the New Mongolia-Siberia DataBase (NMSDB).

| Label | Lat. | Long. | Elev. | Pollen | GDGT | Type | Biomes |
|---|---|---|---|---|---|---|---|
| MMNT5C01 | 48.4074 | 101.8797 | 1433 | x | x | Sediment | Alpine meadow |
| MMNT5C03 | 48.6592 | 101.2015 | 1579 | x | x | Sediment | Alpine meadow |
| MMNT5S04 | 48.4136 | 102.2389 | 1566 | x | x | Sediment | Steppe |
| MMNT5M05 | 48.4203 | 102.2266 | 1538 | x | x | Moss | Steppe |
| MMNT5M06 | 47.7340 | 101.2459 | 1646 | x | | Moss | Steppe-forest |
| MMNT5M07 | 47.7338 | 101.2460 | 1647 | x | | Moss | Steppe-forest |
| MMNT5C11 | 48.9290 | 101.9588 | 1316 | x | | Sediment | Steppe-forest |
| MMNT5C12 | 48.6907 | 101.4263 | 1436 | x | x | Sediment | Steppe-forest |
| MRUT1M01 | 52.0497 | 104.1132 | 565 | x | x | Moss | Dark taiga |
| MRUT1M02 | 52.0498 | 104.1137 | 574 | x | x | Moss | Dark taiga |
| MRUT1M03 | 52.0493 | 104.1132 | 582 | x | x | Moss | Light taiga |
| MRUT1M04 | 52.0500 | 104.1140 | 557 | x | x | Moss | Dark taiga |
| MRUT1M05 | 52.0328 | 104.2263 | 640 | x | x | Moss | Light taiga |
| MRUT1M06 | 52.0148 | 104.2612 | 554 | x | x | Moss | Dark taiga |
| MRUT1M07 | 52.0046 | 104.3738 | 476 | x | x | Moss | Dark taiga |
| MRUT1M08 | 51.9952 | 104.4023 | 471 | x | x | Moss | Light taiga |
| MRUT1M09 | 51.9900 | 104.4025 | 473 | x | x | Moss | Dark taiga |
| MRUT1M10 | 51.9392 | 104.4636 | 538 | x | x | Moss | Light taiga |
| MRUT1M11 | 51.9119 | 104.5331 | 725 | x | x | Moss | Light taiga |
| MRUT1M12 | 51.8797 | 104.6266 | 772 | x | x | Moss | Light taiga |





**Table A3.** Synthesis of the formulae for the main indexes br–GDGT fractional abundances.

| Indice | Formula | Proxy purpose | Reference |
|---|---|---|---|
| MBT | $= \dfrac{\mathrm{Ia+Ib+Ic}}{\sum \mathrm{brgdgt}}$ | Temperature, Precipitation, pH | Weijers et al. (2007b), Huguet et al. (2013) |
| MBT$'$ | $= \dfrac{\mathrm{Ia+Ib+Ic}}{\mathrm{Ia+Ib+Ic+IIa'+IIa''+IIa'''+IIb'+IIb''+IIc'+IIc''+IIIa'+IIIa''+IIIb'+IIIb''+IIIc'}}$ | Temperature, Precipitation | Peterse et al. (2012) |
| MBT$'_{\mathrm{5Me}}$ | $= \dfrac{\mathrm{Ia+Ib+Ic}}{\mathrm{Ia+Ib+Ic+IIa+IIb+IIc+IIIa}}$ | Temperature, Precipitation | De Jonge et al. (2014b) |
| CBT | $= -\log\left(\dfrac{\mathrm{Ib+IIIb+IIIb'}}{\mathrm{Ia+IIa+IIa'}}\right)$ | pH | Weijers et al. (2007b) |
| CBT$_{\mathrm{5Me}}$ | $= -\log\left(\dfrac{\mathrm{Ib+IIb}}{\mathrm{Ia+IIa}}\right)$ | pH | De Jonge et al. (2014b) |
| CBT$'$ | $= -\log\left(\dfrac{\mathrm{Ic+IIa'+IIb'+IIc'+IIIa'+IIIb'+IIIc'}}{\mathrm{Ia+IIa+IIIa}}\right)$ | pH | De Jonge et al. (2014b) |
| R$_{1/b}$ | $= \dfrac{\mathrm{IV+IV'+V+VI+VII+VIII}}{\mathrm{Ia+Ib+Ic+IIa+IIb+IIc+IIIa+IIIb+IIIa}}$ | Aridity | Xie et al. (2012), Yang et al. (2014) |
| BIT | $= \dfrac{\mathrm{Ia+IIa+IIIa+IIIa'+IIIa''+IIIa'''}}{\mathrm{Ia+IIa+IIIa+IIIa'+IIIa''+IIIa'''+IIIa''''+IIa''+IIIa'''+IIIa''''+Cren}}$ | Lake / Basin OM$^b$ production | Hopmans et al. (2004) |
| II/III | $= \dfrac{\mathrm{IIa}}{\mathrm{IIIa}}$ | Lake / Basin OM production | Xiao et al. (2016) |

**Table A4.** a

Climatic and environmental parameters influencing the values of the ratios are presented in the third column. bOM : Organic Matter.



**Figure A1. A** : Cross-plot for $\mathrm{MAP_{mr}}$ model. The parcels are sorted by increasing parameter number. **B** : Cross-plot for selected climate MBT–CBT models.