# Peer review of "Climate reconstructions based on GDGTs and pollen surface datasets from Mongolia and Siberia: Calibrations and applicability to extremely dry and cold environments"

_Biogeosciences, 2019_

## Referee Comment (RC1) · Anonymous Referee #1 · 23 Mar 2020

I have finished a review on the manuscript "Climate reconstructions based on GDGTs and pollen surface datasets from Mongolia and Siberia: Calibrations and applicability to extremely dry and cold environments", submitted to biogeosciences discussions by Lucas Dugerdil and co-authors.

The paper represents an interesting dataset, and great care is taken to develop models based on two different independent paleoclimate data: pollen and brGDGT lipids. I have few comments to make that will hopefully make the manuscript even more relevant for readers interested in using pollen and GDGTs in cold areas. Investigating

[Figure]

Mongolian brGDGT lipid and pollen distributions illustrates local dependencies that are different from global dependencies.

To frame the impact of this observation, the authors need to be more transparent whether and why current global and regional calibrations fail in Mongolia. I would suggest the authors to move the section 5.4. "Issues in Modelling Mongolian Extreme Bioclimate" forward, as this section highlights why dry and cold areas represent challenging conditions for current proxy calibrations. It also allows to present some of the limitations of this study (existing error on climate parameters) before diving into technical model selection steps. For brGDGTs I would recommend to include a figure that shows the variation of MBT'5ME and MAT of soils on a global scale, with the Mongolian soils plotted in this dataset. If the Mongolian soils plot within the global variation of soils, the observed environmental dependencies might be extrapolated to other cold areas. The calibration between currently used brGDGT based proxy values and MAT is mentioned to be plotted in Supp Fig. 3, but this Fig. was not available online.

I am however skeptical of using a heterogeneous brGDGT dataset (including soils, lake sediment data, moss) for developing a new model to predict temperature. Generally, brGDGT from lake sediments are distinct from the surrounding soils, and their different environmental dependency has resulted in the development of lake-specific and soil-specific calibrations. However, in this dataset lake and soil samples are treated as having the same environmental dependency. The MAP is expected to have the same impact on lake-derived brGDGTs as on soils, which makes little sense as there is no link between MAP and aridity in lakes. If the authors have the opinion that the brGDGTs in the lake surface samples are soil-derived and represent an average catchment signal, the authors should elaborate on that. Also, the moss samples are generally not used for brGDGT calibration. Do we expect the brGDGTs to be produced by bacteria within the moss, or to represent an average of soil-derived particles delivered by wind?

The authors mention the effect of aridity on brGDGTs (Dang et al., 2016), but don't include in the discussion whether they see the same brGDGT response in the Mongolian

datasets. The authors have not attempted to make a model to predict MAT, that uses the MAP values as a confounding factor. Why not? The authors state that MAP and MAT correlate in this dataset (this is not plotted anywhere?), but the impact on the models is not discussed. Along the same line: can a partial RDA be used to illustrate the environmental dependencies (MAT or MAP) of brGDGTs in the absence of variation in the second driving factor?

The authors present several models for brGDGT calibration. I agree that caution should be used when using multiple variables (as discussed in section 5.1 and 5.2). Would an adjusted r2 value be useful in this case? I would consider moving the models that are not selected to the supplementary material, so it is clear which model has the best predictive value according to the requirements that the authors have set. Please also see my comment below on selecting a model that makes 'biological sense'.

Comments on content: The introduction introduces several concepts and biomarker lipid groups that are not included in the discussion. Please remove this information (e.g. the impact of human populations on pollen-based climate reconstructions, or the existence of H-shaped or OH-GDGTs). Removing this will open up space to explain those concepts that are used better, and make the relevant part of the introduction more specific.

I also have several minor and 'editorial' comments to make. I indicated 'vague' phrasing where it was difficult for the reader to follow the interpretation:

L 13. Rephrase "derived to the low range of climate parameters encompassed in the study area".

L 19. Rephrase "input proxies" and be more specific. Use fi estimates of climate parameters values. Why does the calibration of the climate proxies matter for the understanding of the interaction between climate model outputs and input? It's a bit arm-wavy, be specific.

L 20. Current climate changes.

L 22. What is meant with 'degradation'?

L25. It is not clear why Mongolia is a hinge area. A hinge area would be influenced by two different climatic drivers, but here the authors state only which climatic drivers do not influence Mongolia. It contradicts L 23-24?

L 29. Perhaps use 'environmental systems' rather then 'climate systems'?

L 31. Please add references here that are an example of the use of pollen and geochemical proxies in lake sediments, or let the sentence reflect the references better.

L 34. Add ' in the absence of human influences'.

L 35. Do you mean specifically the transport mechanism of pollen? 'Pollen rain' is not self-explanatory to me.

L 52-55. This part is too vague. Mention that brGDGTs distributions have been correlated with MAT and pH in soils and in lakes. Please add the reference to Russell et al. (2018) [recent lake calibration that includes temperature and pH]. Refer to Dearing-Crampton Flood as the most recent brGDGT-temperature calibration.

Russell, J.M., Hopmans, E.C., Loomis, S.E., Liang, J., Sinninghe Damsté, J.S., 2018. Distributions of 5- and 6-methyl branched glycerol dialkyl glycerol tetraethers (brGDGTs) in East African lake sediment: Effects of temperature, pH, and new lacustrine paleotemperature calibrations. Organic Geochemistry 117, 56–69.

Dearing Crampton-Flood, E., Tierney, J.E., Peterse, F., Kirkels, F.M.S.A., Sinninghe Damsté, J.S., 2020. BayMBT: A Bayesian calibration model for branched glycerol dialkyl glycerol tetraethers in soils and peats. Geochimica et Cosmochimica Acta 268, 142–159.

L 59. What do the BIT index and the IIIa/IIa index reconstruct? It is not clear what is meant with 'the source'. L 61. Use bacterial producers instead of 'organisms'.

L 63. Better phrase as: 'composition of bacterial community' and refer to De Jonge et al. (2019).

De Jonge, C., Radujković, D., Sigurdsson, B.D., Weedon, J.T., Janssens, I., Peterse, F., 2019. Lipid biomarker temperature proxy responds to abrupt shift in the bacterial community composition in geothermally heated soils. Organic Geochemistry S0146638019301275.

L 64. Please be specific 'climate and soil parameters' could mean many different things!

L 67-68. Are these ratios going to be used in the discussion? If not, consider emitting them.

L 69. Please add what the Ri/b index is (isoprenoid versus branched GDGTs?). How is aridity expected to influence brGDGT distributions?

L 70. Climatic features: please specify which ones.

L 71. The sentence that describes what will be done in this study seems to come too early, perhaps it can be moved to L 79 or removed?

L 73. Rephrase, is there a verb missing here? Based on pollen for instance?

L 77. Help the reader situate this, how many kilometers is there between your site and the other studies?

L 79, multi-proxy calibration comparison study. I.e., you will make a calibration based on more than 1 proxy? Which ones? Perhaps 76-79 can be removed as the following paragraph explains what will happen in this study?

L 84. Please include whether these 'surface samples' are soils, sediments?

L 85. Here you use 'biomarker', next line 'GDGTs'. Please use the same term throughout and be as specific as possible.

L 95. There is no need to add 'in latitude' and 'in longitude'.

L 104. What do the authors mean with "actual GDGT analysis following the calibration approaches presented in De Jonge et al. (2014a), Davtian et al. (2016) 105 and Naafs et al. (2017, 2018)." Do they mean concentrations, fractional abundances, or climate proxy ratios?

L 110. Can the authors add the location of the Darkhad basin to the map?

L 115-125. Can the authors add a map with the expanse of the vegetation types? There is an extensive description, but this information can be conveyed much better with a map. The same goes for the extension of the different biomes (L 130-141).

L 139. Reconsider 'stocked onto', is this a common phrase to describe the orographic effect?

L 155. Please mention again what MAT and WAPLS means here? Reconsider 'Among', for instance: a selection was made from?

L 178. Type: head group.

L 184. Does Davtian et al. (2018) present a modification of the Hopmans et al. (2016) method? Please summarise shortly. If not, remove the reference to Davtian et al. (2018).

L 186. Please rephrase, " compounds were gathered" does not convey how the CBT and MBT proxies are calculated. Please add a formula [I found the formula's below in a Table, refer to this Table in the text] and change the name (De Jonge et al., 2014 uses the MBT'5ME ratio, not the MBT ratio). The formula presented for the fractional abundance is not clear (what is brǎŤGDGT?). The authors can simply state that the fractional abundance of brGDGTs is the % brGDGTs compared to the total of brGDGTs. The same is valid for iGDGTs (iGDGTs/sum(iGDGTs)).

L 192. Say "SIG Bioclimatic Data" instead of "climate parameters", otherwise it is

necessary to specific which climate parameters are meant here.

L 195. Rephrase "gathered some".

L 218. Please explain "AP" and "NAP".

L 245. Specify "geographic scales".

L 249. Rephrase. What is meant with "in case the highest R2 and the lowest RMSE were not applied for the same number of analogues?"

L 251. What do the authors means with "the method fits well".

L 253. What is meant with DB? Database?

L 260. To simply the comparison.

L 267. Replace X. Please add more reference on iGDGTs in soils, it is too simplistic to say that they are only produced in the water column (as they are obviously also present in soils).

L 274: Are consistent between sample types. But are they, FIg. 4B seems to show some differences between samples types? This is also what we would expect from previous studies where a different distribution between soils and lakes was observed?

L 275. Check phrasing.

L 275. If the PCA only explains 37% of the variance within the brGDGTs, is it still useful to show it?

L 289. Which climate parameters? All of the ones mentioned in methods section on SIG Bioclimatic Data?

L 290. I am worried that including surface sediment, soil and moss samples might be introducing variation in the brGDGT dataset, that results in the absence of a correlation between the MBT'5Me and the MAAT. The way forward here would be to split the dataset between lake and soil samples, rather than to go forward and make a new

model. In the PCA I can't see which samples are soils, moss or lake sediment, so it is not clear which impact the different sample types can have on the calibration.

L 294. Were 7-Me brGDGTs encountered in soils? I have not seen any reports of this, are the authors sure this was not a heavy isomer of a brGDGT IIIb that can be seen in the m/z of 1050?

L 305. The authors discuss a statistical approach to select the best model. However, what is neglected is the 'biomechanical logic" behind a model. If the authors follow the reasoning in the introduction, you would expect an increase in branched compounds with decreasing temperature (this is also seen in the global temperature dependencies). Is this reflected by the environmental dependencies in this study? Is this reflected by the models selected?

L 323. How different are these models with different parameters? Can the authors define a requirement and select the best model according to this requierements? And move the rest of the models to Supp. Information?

L 328. (Fig. 7A1).

L 331.Please rephrase, a wide dataset extension is not something that I know of. Also, what are "alleviated extreme values"?

L 342. Is the 2019 paper still in press?

L 349. It would be good to have these results in a figure. Does this concern only soils (for which the soil calibration dataset was developed) or also other sample types?

---

## Referee Comment (RC2) · David Naafs (Referee) · 27 Mar 2020

In this manuscript Dugerdil and co-authors determine pollen and GDGT distributions in a set of surface samples (mineral soil, moss, and lake sediment) from Mongolia and Siberia to establish novel (local) climatic calibrations (precipitation and temperature).

I am not an expert in pollen and therefore focus my review on the GDGT-part of the manuscript. I hope other reviewers can comment on the pollen methods and results. I congratulate the authors with writing a manuscript that reads well and with references

that are up to date. The figures are also well made. I especially like figure 7 that manages to clearly display a lot of information (but check if it is suitable for colourblind people?). I think this is an elegant dataset from a poorly studied region that is valuable to the paleoclimate, pollen, and GDGT community. However, I have a number of comments that need to be addressed before publication.

- Firstly, the foundation for the calibrations: the instrumental data. Given the few weather stations in the region (line 169), what is the error on the instrumental data (temperature and precipitation) for the calibrations? For example, for temperature is it 1 or 5 oC or more? How confident are we that we know the temperature that the samples experienced? This error should be taken into account into the calibrations and discussed properly. Also, the discussion focusses on mean annual temperature, but did you explore warm season temperatures? Especially for the cold regions this might improve the calibration. If not, this is also interesting and should be discussed.

-I find it odd that lake sediment samples are combined with mineral soil and moss samples. Does that mean the new calibrations can be applied to both archives? We know that the brGDGT distribution in lakes differs with that in soils (compare mineral soil brGDGT data versus temperature with that of lakes). I suggest the authors split the lake and soil data into different calibrations. Does that improve the correlation for for example MBT5me'? Also compare the lake samples with the recent lake calibrations (Russell et al., 2018). In addition, for lake sediments I expect no correlation between brGDGT distribution and local precipitation (as they are mainly formed in the water column), was this taken into account to obtain the MAP calibrations?

-But then there is the more fundamental problem: Using R2 values and other statistical methods to select the best calibration. I appreciate that the authors take overparametrization into account (section 5.1) but I think a major problem with their approach is that we end up with complex calibrations that include compounds that are not very abundant (e.g. brGDGT-IIIa' and -IIIc). Minor changes in the abundance of these minor compounds (or even slightly different ways of integrating the minor peaks)

can have a major impact on the resulting temperature. I think it would be valuable to only consider compounds that have a certain relative abundance, like > 5-10 %. How would this impact the selected calibrations? In addition, this way we lose any physical basis for the proxy. The original MBT proxy reflects a decrease in degree of methylation with increasing temperature and this has a physical meaning for membrane properties and hence provides confidence that this is a true signal and not a random empirical observation. But what is the impact on a bacterial membrane if it is has a few percent more of brGDGT-IIIb'? What is the physical basis of these new calibrations and hence are we confident that these are real temperature relationships? This aspect needs a thorough consideration and discussion.

-Lastly, what is missing from this manuscript is an application of these novel calibrations. Do they provide sensible climatic signals when applied to a downcore record? If the authors do not have a downcore record, is there a published record that this calibration can be applied to?

David Naafs

Minor comments:

Line 6: delete "extremely cold dry"

Line 7: is the livestock grazing statement relevant? It does not appear to come back in the discussion, delete?

Line 16: add "environmental variations in Mongolia and Siberia"

Line 52 and 55: brGDGTs

Line 55: also cite other recent brGDGT calibrations papers, for example (Wang et al., 2019)

Line 62-63: I don't think we show this in our 2018 paper. Delete sentence or change reference

Introduction needs some rewriting to create a more natural flow between the different paragraphs. At the moment the introduction jumps from pollen to brGDGTs and back without a clear connection between the different paragraphs. A good example is the sentence in Line 71 that sort of floats by itself with no connection to the previous sentences or the next paragraph. The end of the introduction with a clear outline of the approach is good. Also, I think somewhere in the introduction the authors need to introduce the 5,6,7 methyl brGDGTs and what they mean (For example citing De Jonge et al., 2013; Ding et al., 2016). Because at the moment their use (e.g. line 276) comes a bit out of nowhere (need also somewhere in the methods explain the use of ' and '' for 6, and 7-methyl brGDGTs).

Line 265-270: this section needs a bit of re-writing (and maybe thinking). We know isoGDGTs are also produced in soils and peats, not just in lakes and not only in the water column. So that is not the reason that you see no clear correlation with climate parameters. See the discussion in (the supplementary information of) (Naafs et al., 2018) on the distribution of isoGDGTs in peat and the lack of correlation with temperature/pH. It is interesting that in these samples from dry environments crenarchaeol is abundant because the abundance of crenarchaeol in soils/peat has been inferred to indicate dry conditions (see for example Zheng et al., 2015), I suggest the authors expand on this.

Line 274: I don't understand this sentence

Figure 4: change legend to state "lake sediments, n=65"

Line 279: what kind of surface samples? Soil surface?

Line 297-298: On what basis were these 9 selected? Expand

Section 5.3: I think the assumption for this section is flawed. Of course, a local calibration based on a certain set of surface samples will have a better correlation with the MAAT then a global calibration applied to these surface samples. You chose your local

calibration using these same samples.

Line 379: also see discussion in (De Jonge et al., 2014; Naafs et al., 2017) on brGDGTs in dry soils

References

De Jonge, C., Hopmans, E.C., Stadnitskaia, A., Rijpstra, W.I.C., Hofland, R., Tegelaar, E., Sinninghe Damsté, J.S., 2013. Identification of novel penta- and hexamethylated branched glycerol dialkyl glycerol tetraethers in peat using HPLC–MS2, GC–MS and GC–SMB-MS. Organic Geochemistry 54, 78-82, doi: 10.1016/j.orggeochem.2012.10.004

De Jonge, C., Hopmans, E.C., Zell, C.I., Kim, J.-H., Schouten, S., Sinninghe Damsté, J.S., 2014. Occurrence and abundance of 6-methyl branched glycerol dialkyl glycerol tetraethers in soils: implications for palaeoclimate reconstruction. Geochimica et Cosmochimica Acta 141, 97-112, doi: 10.1016/j.gca.2014.06.013

Ding, S., Schwab, V.F., Ueberschaar, N., Roth, V.-N., Lange, M., Xu, Y., Gleixner, G., Pohnert, G., 2016. Identification of novel 7-methyl and cyclopentanyl branched glycerol dialkyl glycerol tetraethers in lake sediments. Organic Geochemistry 102, 52-58, doi: 10.1016/j.orggeochem.2016.09.009

Naafs, B.D.A., Gallego-Sala, A.V., Inglis, G.N., Pancost, R.D., 2017. Refining the global branched glycerol dialkyl glycerol tetraether (brGDGT) soil temperature calibration. Organic Geochemistry 106, 48-56, doi: 10.1016/j.orggeochem.2017.01.009

Naafs, B.D.A., Rohrssen, M., Inglis, G.N., Lähteenoja, O., Feakins, S.J., Collinson, M.E., Kennedy, E.M., Singh, P.K., et al., 2018. High temperatures in the terrestrial mid-latitudes during the early Paleogene. Nature Geoscience 11, 766-771, doi: 10.1038/s41561-018-0199-0

Russell, J.M., Hopmans, E.C., Loomis, S.E., Liang, J., Sinninghe Damsté, J.S., 2018. Distributions of 5- and 6-methyl branched glycerol dialkyl glycerol tetraethers

(brGDGTs) in East African lake sediment: Effects of temperature, pH, and new lacustrine paleotemperature calibrations. Organic Geochemistry 117, 56-69, doi: 10.1016/j.orggeochem.2017.12.003

Wang, M., Zheng, Z., Zong, Y., Man, M., Tian, L., 2019. Distributions of soil branched glycerol dialkyl glycerol tetraethers from different climate regions of China. Scientific Reports 9, 2761, doi: 10.1038/s41598-019-39147-9

Zheng, Y., Li, Q., Wang, Z., Naafs, B.D.A., Yu, X., Pancost, R.D., 2015. Peatland GDGT records of Holocene climatic and biogeochemical responses to the Asian Monsoon. Organic Geochemistry 87, 86-95, doi: 10.1016/j.orggeochem.2015.07.012

―――――――――――――――――――

---

## Author Comment (AC1) · 7 May 2020

[12pt]article

We thank the reviewers for their attentive reading and their accurate comments. We certainly appreciate the feedback they provided, and have strove to improve our manuscript according to their suggestions. We also provide a point-by-point account of our rebuttal, please see below. In addition to the changes suggested by the referees, we also modified the following items:

- 1. The annexes have been reorganized and the references in text to tables and figures have been updated and sorted.

- 2. Some recent references have been added to reinforce our assumptions.

- 3. The formula of the MAAT - brGDGT calibration used in Figure 7 was added in the Table Annex A3.

**Responses to the comments of Reviewer 1 (Anonymous Referee)**

General comments:

I have finished a review on the manuscript " *Climate reconstructions based on GDGTs and pollen surface datasets from Mongolia and Siberia: Calibrations and applicability to extremely dry and cold environments* ", submitted to biogeosciences discussions by Lucas Dugerdil and co-authors. The paper represents an interesting dataset, and great care is taken to develop models based on two different independent paleoclimate data: pollen and brGDGT lipids.

I have few comments to make that will hopefully make the manuscript even more relevant for readers interested in using pollen and GDGTs in cold areas. Investigating Mongolian brGDGT lipid and pollen distributions illustrates local dependencies that are different from global dependencies. To frame the impact of this observation, the authors need to be more transparent whether and why current global and regional calibrations fail in Mongolia.

**Response**: Local and global dependencies are actually not totally disconnected from each other. The Sibero-Mongolian system is indeed driven by the global climatic system, elevation gradients and its ecological responses. However, because of the extreme values of all these parameters, the local dependency has to be investigated in details. Basically, if the global calibration could be used in palaeoclimatic reconstructions for the Sibero-Mongolian lakes, the accuracy and the interpretation of the results has to be questioned (cf. Figure 7).

**Applied changes**: To emphasize the limits of local applications of global calibrations, we have added a sentence at the end of the Global vs. Local Calibration paragraph : " *Tangibly, for the two proxies, even if the global calibrations can operate on our study area, the local calibrations reach higher accuracy* " (L. 401). Moreover the following sentence has been added in conclusion. " *Even if global and regional calibrations could be applied in such a setting, local calibrations provide enhanced accuracy and specificity.* " have been introduce within the conclusion. (L. 485)

I would suggest the authors to move the section 5.4. " *Issues in Modelling Mongolian Extreme Bioclimate* " forward, as this section highlights why dry and cold areas represent challenging conditions for current proxy calibrations. It also allows to present some of the limitations of this study (existing error on climate parameters) before diving into technical model selection steps.

**Response**: We acknowledge this remark but we rather prefer to discuss the details and technical limits of our methodology first and then to extend these limits latter to the global context of Mongolian climatic system and to global palaeo-proxy calibrations. We therefore choose, not to modify the structure of the discussion. About the existing error on climate parameters, we add information (following a comment in the review of D. Naafs)

**Applied changes**: paragraph 5.4 " *According to the authors, the interpolation model*

*used in the Central Arid Area (which include our study area) gives $R^2$ = 0.99 and RMSE = 1.3 C for MAAT and $R^2$ = 0.89 and RMSE = 23 mm.yr-1 for MAP. Whenever the Siberian-Mongolian calibrations are used for palaeoclimatic reconstructions, the RMSE of the climate parameter have to be added to the model RMSE.* " (L. 416)

For brGDGTs I would recommend to include a figure that shows the variation of MBT'5ME and MAT of soils on a global scale, with the Mongolian soils plotted in this dataset. If the Mongolian soils plot within the global variation of soils, the observed environmental dependencies might be extrapolated to other cold areas.

**Response and applied changes**: In order to test whether our dataset was included in the world trends for MBT and MBT'5ME–MAAT relationships, we plotted these graphs in an early stage of our investigations based on the global and regional chinese databases (Naaf et al., 2017 - Dearing Crampton-Flood et al., 2019 and Yang et al., 2014, respectively). On both panels A and B, our dataset plot within the global trend, but the vertical dispersion is larger than the horizontal one. This leads us to the conclusion that MBT and MBT-derived indexes are hardly applicable for local climatic calibrations. As suggested by the reviewer 1 we added this new figure in the Annex B2, here Fig. 1 in the interactive discussion.

The calibration between currently used brGDGT based proxy values and MAT is mentioned to be plotted in Supp Fig. 3, but this Fig. was not available online.

**Response and applied changes**: The reference was unfortunately mislabeled. The reference has been updated (Figure B3).

I am however skeptical of using a heterogeneous brGDGT dataset (including soils, lake

sediment data, moss) for developing a new model to predict temperature. Generally, brGDGT from lake sediments are distinct from the surrounding soils, and their different environmental dependency has resulted in the development of lake-specific and soil-specific calibrations. However, in this dataset lake and soil samples are treated as having the same environmental dependency. The MAP is expected to have the same impact on lake-derived brGDGTs as on soils, which makes little sense as there is no link between MAP and aridity in lakes. If the authors have the opinion that the brGDGTs in the lake surface samples are soil-derived and represent an average catchment signal, the authors should elaborate on that.

**Response :** The second reviewer raised a similar concern. Actually the calibration is mainly based upon moss and soil samples (or 98%), only 4 sediment samples were considered for pollen and 3 for br-GDGTs. Cross-values were used to check the response to all the models (the one from the literature as well as the models from this study). A sample has to be totally independent of the calibration to be unbiased, that is why MMNT5C12 was removed from the br-GDGT calibration database. The 2 others sediment samples are the lake shores, not top-cores. They are made of humid clay with desiccation cracks and embedded peat organic matter, therefore a different feature of lacustrine sediments.

**Applied changes :** To take this comment into account and clarify this concern, we have modified the figure 4 to highlight that the majority of the sediment samples are independent of the sample NMSDB (just displayed as a comparison). The caption of the Figure 7 has also been modified with " *cross-value on the 6 first samples of the independent core MMNT5C12, Arkhangai* ". We furthermore added in the 3.4 Statistical Analyses paragraph the following sentence: " *A cross-value test was performed for all the models, the previous studies as well as the models proposed in this study, using an independent set of the six first sediment core samples from*

*the lake MMNT5C12, Arkhangai* " (L. 218) In the result part, we also observe no influence of the lake system in iso-GDGT relative abundances: part 4.2.1 (L. 281): " *Iso-GDGT pattern in lake sediments do not really diverge from surface samples which leads to postulate that the in-situ production of iso-GDGTs in shallow lakes like MMNT5C12 is reduced (Fig. 4.A).* " Finally the discussion part has been extended (L. 455) : " *Moreover, in the desert context of poor availability in archive sites, the edge clay samples or top-cores of shallow lakes could be a solution. The two sediment samples of NMSDB are in the soil-moss trend for all models (Fig. 5, Annex Fig. B2 and B3). Even if the br-GDGT production and concentration differ in soils and in lakes due to in-situ production (Tierney and Russell, 2009 ; Buckles et al., 2014), this effect is function of the lake depth (Colcord et al., 2015), consequently seems to be negligible for shallow lakes, and almost absent in lake edge samples (Coffinet, 2015).* "

Also, the moss samples are generally not used for brGDGT calibration. Do we expect the brGDGTs to be produced by bacteria within the moss, or to represent an average of soil-derived particles delivered by wind?

**Response** : Moss samples are indeed not usually used in br-GDGT calibrations, but our methodology was to develop a common protocol for pollen and br-GDGT proxies. Because in many cases the moss polsters are better for pollen surface samples than soils (less taphonomic biases for example), this type of samples is preferred. On that purpose, we wished to assess if the br-GDGT pattern and abundances would be impacted by sample type (as it is shown in Figure 4). Our conclusion is that whenever the choice is possible in the field, the best option is to take parallel samples of soils and moss pollsters for br-GDGTs and pollen, respectively. In fact, on the Annex Figure A2, we observe that the moss samples plot within the peat data set (Naafs, 2017), while the soil samples are mostly associated with Chinese soil trend confirming the reviewer comment.

**Applied changes** : (L. 183) " *In order to compare with the methodologies developed for pollen, moss polsters, soil samples and two lake shore sediment samples have been treated for GDGT analysis.* " has been added in the GDGT methods. In Discussion 5.4 : (L. 452) " *The explanation of the signal difference between the two types of samples could also originate from the in-situ production of br-GDGTs inside the moss in front of the wind-derived particles brought to the moss net. On the annexe figure B2.A it seems that the pool of moss polster is associated with with a similar trend that the worldwide peat samples from Naafs (2017).* "

The authors mention the effect of aridity on brGDGTs (Dang et al., 2016), but don't include in the discussion whether they see the same brGDGT response in the Mongolian datasets. The authors have not attempted to make a model to predict MAT, that uses the MAP values as a confounding factor. Why not?

**Response and applied changes**: We agree with the referee's comment. Because the Mongolian plateau is a dry area, all the studies demonstrating that br-GDGT abundances suffer from moisture influence need to be taken into account. Based on the Dang et al., (2016) study, we justify the use br-GDGT abundances as proxy for MAP. Their results on moisture impact on brGDGT responses are discussed along two lines:

- Part 5.4 : the MAP–brGDGT correlation remains strong (Annex Table 2, Bottom Part): so mathematically we observe a positive response of brGDGTs to MAP.

- Secondly, if we cannot discuss the eco-physiological controls of MAP (or even better soil water content) similarly to what is done by Dang et al., (2016), because we lack these observations, we have proposed two models derived from brGDGTs : one for MAAT and the other one for MAP, making the hypothesis

that the two variables are independent. It is actually not true but the specific responses of brGDGT to MAAT or MAP still remain unknown.

The authors state that MAP and MAT correlate in this dataset (this is not plotted anywhere?), but the impact on the models is not discussed. Along the same line: can a partial RDA be used to illustrate the environmental dependencies (MAT or MAP) of brGDGTs in the absence of variation in the second driving factor?

**Response** : MAP and MAAT are indeed not correlated in the same way in all the NMSDB samples as illustrated below (added in Annex Figure B1, on interactive comments fig. 2). In the present form of the manuscript, we discuss the difficulties in modeling the Mongolian dry plateau, but on the total range of sample sites, the $R^2$ is only 0.35. On the other hand, on the subset of the Mongolian sites, $R^2$ value reaches 0.91, which shows a strong correlation between the two variables. This tendency remains similar along the latitude gradient: decreasing precipitations is linked to rising temperatures. If the auto-correlation between MAAT and MAP represents a risk for the models reliability, because or data set employed both Siberian and Mongolian sites ($R^2$ = 0.35) the models stay reliable. Concerning the RDA " *br-GDGT variations linked to environmental factors* " shows a RDA1 component mainly explained by a gradient between cold-wet Siberian forests against hot-dry Gobi Desert. The second parameter (RDA2) is the altitude gradient introducing variability on the first main axis. On a biological point of view, the altitude could maybe induce a modification in bacterial communities, in their response and sensibility to climatic variations, or even a shift in the bacterial-vegetation relationship. The vegetation is indeed forced by the elevation gradient (not only because of temperature and precipitation changes but also because of the exposure, slop and soil creeping, wind intensity and 02 concentration).

**Applied changes** : Figure A3 has been added in Annex and the paragraph 5.4 has been modified with the addition of (L. 412): " *Issues in Modelling Mongolian Extreme Bioclimate* " and at the end of the paragraph (L. 421) " *Because the MR–GDGT models have been compiled with the group of Siberian sites which are out of the MAAT–MAP strong auto-correlation (annexe fig. B1) the reliability of the independence of the MAAT and MAP models is guaranteed.* " On the reliance of the elevation underlined by the RDA, the discussion (part 5.4) was modified (L. 424): " *Elevation as main br-GDGT drivers could also be explained by the bacterial community responses to pH, moisture and soil compound variations along the altitude gradient (Laldinthar and Dkhar, 2015; Shen et al., 2013; Wang et al., 2015) and the vegetation shifts (Lin et al., 2015; Davtian et al., 2016; Liang et al., 2019).* "

The authors present several models for brGDGT calibration. I agree that caution should be used when using multiple variables (as discussed in section 5.1 and 5.2). Would an adjusted $R^2$ value be useful in this case?

**Response** : We fully agree with the reviewer's comment, the adjusted $R^2$ is indeed particularly adapted for multiple linear regression models because it allows to check the level of accuracy of the models in one hand and to reduce the impact of the over-paramerization on the other hand. For this study, we however choose to use a combination of $R^2$ + AIC because (1) it leads to the same statistical control (accuracy + limitation of over-parametrization) and (2) it allows to simultaneously evaluate the effect of the two phenomena for each mode. Moreover, because the vast majority of the paleoclimate proxy calibration studies used the $R^2$, it appear more convenient to use the same statistical parameters for immediate comparison. When specifically tested, the use of the adjusted-$R^2$ provided similar results than $R^2$ + AIC. For instance, on a set of models, the best adjusted- $R^2$ model is also the model with the smallest AIC.

I would consider moving the models that are not selected to the supplementary material, so it is clear which model has the best predictive value according to the requirements that the authors have set.

**Response**: We understand the reviewer's concern to improve the readiness of the manuscript. However, One of our main issues is to help model-makers to suggest a methodological path for best model selections. We thing it might be meaningful to display some of the imperfect models to discuss the way of assessing their usefulness through their formulas and statistic values. Moreover, we wish to emphasize that the model selection is context-dependent.

**Applied changes** : To simplify the take-home message, the best model for Mongolia (in blue in Table 2) and the best model for the " *cold and dry similar environment* " (in red in Table 2) will be spotlighted by colors in Table 2. We reformulated Paragraph 5.1 (L.345) " *To sum-up, the most universal models are MAAT mr5 and MAP mr7 (Table 2, red coloured models) but the closed models are also valuable in some local contexts, and especially in similar dry-cold regions* . "

Please also see my comment below on selecting a model that makes " biological sense ". Comments on content: The introduction introduces several concepts and biomarker lipid groups that are not included in the discussion. Please remove this information (e.g. the impact of human populations on pollen-based climate reconstructions, or the existence of H-shaped or OH-GDGTs). Removing this will open up space to explain those concepts that are used better, and make the relevant part of the introduction more specific.

**Response and applied changes**: We modified accordingly the text:

- even if the H-GDGT and the OH-GDGT were analyzed in the NMSDB, references to these compounds were removed from the introduction.

- similarly, BIT and IIIa/IIa indexes were removed as well as referring references.

Specific comments:

I also have several minor and " editorial " comments to make. I indicated 'vague' phrasing where it was difficult for the reader to follow the interpretation: L 13. Rephrase " *derived to the low range of climate parameters encompassed in the study area* ".

**Response and applied changes:** We have modified point (3) of the introduction by " *Even if local calibrations suffer from reduced climatic parameter amplitudes due to local homogeneity, they better reconstruct climatic parameters than the global ones by reducing the limits for saturation impact* " (L.11)

L 19. Rephrase " *input proxies* " and be more specific. Use fi estimates of climate parameters values. Why does the calibration of the climate proxies matter for the understanding of the interaction between climate model outputs and input? It's a bit arm-wavy, be specific.

**Response and applied changes:** The precision " *such as pollen or biomarker abundances* " to explain the " *input proxies* " was added (L.18). The more accurate and specific the calibration, the more we can understand on climate reconstructions by models.

L 20. Current climate changes.

**Response and applied changes:** Modified accordingly.

L 22. What is meant with " *degradation* " ?

**Response and applied changes**: According to Tian et al., (2014) this means the " *diversity and biomass production* " degradation. The sentence was added to the manuscript.

L25. It is not clear why Mongolia is a hinge area. A hinge area would be influenced by two different climatic drivers, but here the authors state only which climatic drivers do not influence Mongolia. It contradicts L 23-24?

**Response and applied changes**: In an attempt to better describe the specificity of the climate controls on Mongolia, we have modified the sentence by removing " *partially block the Westerlies...* " and concluded by adding " *The Mongolian system is thus driven by a mix of the distant drags of these two climatic cells.*"

L 29. Perhaps use " *environmental systems* " rather then " *climate systems* " ?

**Response and applied changes**: Modified accordingly: we have changed the sentence by " *climate and environmental systems* "

L 31. Please add references here that are an example of the use of pollen and geochemical proxies in lake sediments, or let the sentence reflect the references better.

**Response and applied changes:** To highlight the combined applications of these proxies we have modified the sentence by adding " *and more in a comparative attempt on lake sediment environmental interpretations* " and add the references to Atahan et al., 2015; Watson et al., 2018 and Martin et al., 2019.

L 34. Add " *in the absence of human influences*".

**Response and applied changes:** Modified accordingly.

---

## Author Comment (AC2) · 7 May 2020

**Responses to the comments of Reviewer 2 (David Naafs)**

General comments:

In this manuscript Dugerdil and co-authors determine pollen and GDGT distributions in a set of surface samples (mineral soil, moss, and lake sediment) from Mongolia and

[Figure]

Siberia to establish novel (local) climatic calibrations (precipitation and temperature). I am not an expert in pollen and therefore focus my review on the GDGT-part of the manuscript. I hope other reviewers can comment on the pollen methods and results. I congratulate the authors with writing a manuscript that reads well and with references that are up to date. The figures are also well made. I especially like figure 7 that manages to clearly display a lot of information (but check if it is suitable for colourblind people?). I think this is an elegant dataset from a poorly studied region that is valuable to the paleoclimate, pollen, and GDGT community.

However, I have a number of comments that need to be addressed before publication. - Firstly, the foundation for the calibrations: the instrumental data. Given the few weather stations in the region (line 169), what is the error on the instrumental data (temperature and precipitation) for the calibrations? For example, for temperature is it 1 or 5 C or more? How confident are we that we know the temperature that the samples experienced? This error should be taken into account into the calibrations and discussed properly.

**Response and applied changes**: The climate data originate from the WorldClim2 database. The error on the climatic parameters is linked to elevation and distance to stations. Because these two sets of information, it should possible to provide uncertainties. Such a modeling could however be the topic of a whole article per se, we therefore based our analyses on the errors proposed by Fick and Hijmans (2017) for the Arid Central Asian Area. The interpolation model used f(x,y,z, coast, sat) in this area give a R-squared = 0.994 and RMSE = 1.344. This error was established on a large geographical area, for which elevation and isolation (average distance between two meteorological stations) are basically similar to the Mongolian plateau. Similar reasoning was followed for MAP : $R^2$ = 0.894 and RMSE = 23.241 mm.yr-1. We mentioned these errors in the manuscript (Paragraph 5.4 followed, L.

419) and added the following sentence: "*According to the authors, the interpolation model used in the Central Arid Area (which includes our study area) gives $R^2$ = 0.99 and RMSE = 1.3 C for MAAT and $R^2$ = 0.89 and RMSE = 23 $mm.yr^{-1}$ for MAP. Whenever the Siberian-Mongolian calibrations are used for palaeoclimatic reconstructions, the RMSE of the climate parameters have to be added to the RMSE model.*"

Also, the discussion focuses on mean annual temperature, but did you explore warm season temperatures? Especially for the cold regions this might improve the calibration. If not, this is also interesting and should be discussed.

**Response:** For pollen transfer functions, the Mean Temperature of the Warm Period (MTWA) or spring temperatures seem to better fit than the MAAT on siberian-mongolian clipped databases. Similarly, for br-GDGTs, MR-models have also been calculated with MTWA and summer temperatures. These models do not however perform significatively better than those with MAAT. For example, the MTWA$_{mr6}$ (6 parameters) presents $R^2$ = 0.63, RMSE = 1.53C and AIC = 178. As summer temperatures are concerned, MTWA$_{mr6}$ displays the exact same values (probably because MTWA and Tsum are roughly spreading on the same time-laps). Similarly, MTWA do not perform significantly better than MAAT mr-models. Although this lack of seasonality is quite surprising (because of the permafrost impact on bacterial communities), this result is consistent with the Chinese sites not showing any seasonal bias (Lei et al., 2016).

Applied changes : At the end of the second part of the 5.4 paragraph (L. 445) : "*Even if GDGTs seem to react to summer temperature, (Wang et al., 2016; Kusch et al., 2019) the mr-models are not significantly improving the calibration than the*MAAT$_{mr}$ *ones. For instance, the best* Tsum$_{mr}$ *is selected by its AIC,* Tsum$_{mr6}$ *using 6 br-GDGTs fractional abundance displays $R^2$ = 0.63 and RMSE = 1.53 C. The lack of seasonality,*

*expected in such cold areas, is consistent with temperate Chinese sites (Lei et al., 2016).*"

-I find it odd that lake sediment samples are combined with mineral soil and moss samples. Does that mean the new calibrations can be applied to both archives? We know that the brGDGT distribution in lakes differs with that in soils (compare mineral soil brGDGT data versus temperature with that of lakes). I suggest the authors split the lake and soil data into different calibrations. Does that improve the correlation for for example MBT5me'? Also compare the lake samples with the recent lake calibrations (Russell et al., 2018). In addition, for lake sediments I expect no correlation between brGDGT distribution and local precipitation (as they are mainly formed in the water column), was this taken into account to obtain the MAP calibrations?

**Response:** This concern is important and was similarly raised by Referee 1 (see the response and the associated changes in the answer to Referee 1).

-But then there is the more fundamental problem: Using $R^2$ values and other statistical methods to select the best calibration. I appreciate that the authors take overparametrization into account (section 5.1) but I think a major problem with their approach is that we end up with complex calibrations that include compounds that are not very abundant (e.g. brGDGT-IIIa' and -IIIc). Minor changes in the abundance of these minor compounds (or even slightly different ways of integrating the minor peaks) can have a major impact on the resulting temperature. I think it would be valuable to only consider compounds that have a certain relative abundance, like > 5-10 %. How would this impact the selected calibrations? In addition, this way we lose any physical basis for the proxy. The original MBT proxy reflects a decrease in degree of methylation with increasing temperature and this has a physical meaning for membrane properties and hence provides confidence that this is a true signal and not

a random empirical observation. But what is the impact on a bacterial membrane if it is has a few percent more of brGDGT-IIIb'? What is the physical basis of these new calibrations and hence are we confident that these are real temperature relationships? This aspect needs a thorough consideration and discussion.

**Response and applied changes**: We acknowledge this comment, but we wish to raise several related issues:

- First, the "minor compounds" are not identical depending on the setting. For example, IIIa' and IIIc are respectively the 4th and the 9th compound by relative abundances over the 15 compounds. Furthermore, the compound distributions in the Siberian-Mongolian area differ from the global peatland and soil database (Figure X). In global datasets, the relative abundances are damped because the over representation of Ia (45 to 75 % in peat samples) and IIa (15 to 40 % in peat samples). In mongolian soils, these two compounds are less abundant, 10 to 20 % and 5 to 15%, respectively (Figure X).

- Applied changes: Paragraph 4.2.3 : "Even the models including minor compounds ([br]i<5%) have been studied. In the NMSDB brGDGT fractional abundances are indeed more fairly distributed than in the global database, where some compounds overlaps the others (Annex Figure B2.C and D)"

- Furthermore, when compound cumulative means are considered (Annex fig. B2.D), the curve issued from the Mongolian dataset is flatter than the global peat and soil ones. The impact of minor compounds finally appear more important than in other studies. This is especially the case for some compounds, not abundant in the global context but more expressed in our study such as : [IIIa'], [Iia'],[IIb'] and [IIIc]. More importantly, if the model is forced to select only compounds with [br]i>10% or even 5%, correlations to MAP and MAAT disappear (best $R^2$ around 0.17 and 0.14, respectively).

[Figure]

- Last, the main purpose of this manuscript is to develop a common methodology for both pollen and GDGT modeling using statistical selections of modern analogues and without any eco-physiological consideration. This choice was made based on the lack of constrains on biological processes driving GDGT production. This holds true for pollen, a proxy where the relation between climate – vegetation and pollen production is definitely much restrained. The biologically blind method is thus useful to avoid over-explanations of causes and consequences.

**Applied Change:** To clarify these issues, the 2nd point of the steps at the end of the introduction (L. 94) has been modified in :"Evaluation of the match between actual bioclimate environments and associated pollen rain and biomarker assemblages based on mathematical criterion without any eco-physiological considerations."

- Even if all models presented in this study, MR-GDGT models included, are made on eco-physiological blind test, some physical laws governing these models can be derived.

- First, the main molecules involved in the mr-models are overall matching with the RDA vectors. For instance, the $MAAT_{mr}$ models apply $+ [IIIa']$ and $- [IIIa]$, consistently with the GDGT-climat RDA. This indicates that, on our data-set at least, the mr-models follow the climatic forcing displayed in the RDA analysis. Table Fig.2 (in the interactive comments) presents the 4th mr-GDGT model selected as the better representation of Sibero-mongolian climate. br-GDGT compounds are ordinated by decreasing proportions. Number and sign in cells show the importance of each br-GDGT in the model and if they are positively or negatively used in the model.

- On the table S1, except for [IIb'], all the compounds change their sign when used with another climatic parameter. For instance [IIIa] is positively correlated with

MAAT and negatively with MAP. It is consistent with the anti-correlation between MAP and MAAT (Annex Fig. B1).

- Then, we observe that the main variables for MAAT are gathered in the group G1 mainly composed of penta and hexamethylated brGDGTs, while the MAP in G2 is mainly based on tetra and pentamethylated brGDGTs (table Fig. 2 in the interactive comments). This group G2 is also composed exclusively by compounds with 2 (b) or 3 (c) cycles. The MAP reconstruction is then leaded by the cycled molecules. These componds are known to be linked to pH (Damste et al., 2016, Weijers et al., 2007b) with an increasing number of cycles for basic soils (pH>7). The basic soils are often associated with low precipitation area because of the weak weathering effect (Dregne et al., 1976; Haynes et al., 1989).

**Applied Change:** Paragraph 4.2.3 (L. 324) : "*Both* $MAAT_{mr}$ *models infer on a positive contribution of [III'a] and a negative contribution of [IIIa], which confirm these models are eco-physiologically consistent with RDA results. Moreover, except for [IIb'], all compounds are positively correlated with MAAT and negatively with MAP, in accordance with MAP - MAAT anti-correlation.*" Paragraph 5.4 (L. 410) "*(...) arid soils favor 6-Methyl by pH raising due to the low weathering effect of the weak precipitation (Dregne et al., 1976; Haynes et al., 1989)*"

-Lastly, what is missing from this manuscript is an application of these novel calibrations. Do they provide sensible climatic signals when applied to a downcore record? If the authors do not have a downcore record, is there a published record that this calibration can be applied to?

**Response:** We acknowledge the comment by the referee, the application of these calibrations to a paleorecord is the ultimate goal of this study. We first considered

submitting the two sets of data in the same manuscript but the resulting text was too long to properly address the issues raised by the calibration. We chose to first describe in details the methodology as well as the advantages and pitfalls of statistical treatments for deriving MAAT and MAP calibrations in arid areas. This will allow to document in a second step, the implications of this calibration exercise in terms of paleoclimate reconstructions.

Specific comments:

Line 6: delete "extremely cold dry"

**Response and applied changes**: Modified accordingly

Line 7: is the livestock grazing statement relevant? It does not appear to come back in the discussion, delete?

**Response and applied changes**: Modified accordingly. The grazing is actually particularly relevant for the understanding of alpine and semi-desert vegetation and consequently the pollen rain. However, since we did not focus on Non Pollen Palynomorph (NPP), we did not develop much this point in the present manuscript.

Line 16: add "environmental variations in Mongolia and Siberia"

**Response and applied changes**: Modified accordingly.

Line 52 and 55: brGDGTs

**Response and applied changes**: Modified accordingly.

Line 55: also cite other recent brGDGT calibrations papers, for example (Wang et al., 2019)

**Response and applied changes**: Modified accordingly: Wang et al., 2019 and 2020 on attitudinal transects.

Line 62-63: I don't think we show this in our 2018 paper. Delete sentence or change reference.

**Response and applied changes**: We were referencing to "*Whether this is related to changes in the source organism's brGDGT distribution or due to changes in the bacterial community composition is currently unclear.*" which is not the main subject of the article but raised similar doubts. The sentence has been modified into "*Moreover some studies have focused on the variations in the bacterial community structure (Xie et al., 2015), the bacterial group responses to environmental changes (Knappy et al., 2011) and the GDGT occurrences in different bacterial communities (Liu et al., 2012b) to determine the possible community structure impact of GDGT abundance.*" (L. 65)

Introduction needs some rewriting to create a more natural flow between the different paragraphs. At the moment the introduction jumps from pollen to brGDGTs and back without a clear connection between the different paragraphs. A good example is the sentence in Line 71 that sort of floats by itself with no connection to the previous

sentences or the next paragraph. The end of the introduction with a clear outline of the approach is good.

**Response and applied changes**: We agree with this general comment on the introduction. The link between the paragraphs have been smoothed.

Also, I think somewhere in the introduction the authors need to introduce the 5,6,7 methyl brGDGTs and what they mean (For example citing De Jonge et al., 2013; Ding et al., 2016). Because at the moment their use (e.g. line 276) comes a bit out of nowhere (need also somewhere in the methods explain the use of ' and '' for 6, and 7-methyl brGDGTs).

**Response and applied changes**: The 5th paragraph of the introduction dedicated to GDGTs have been enriched by :"In particular, the methylation degrees, ratios of 5, 6-methyl (De Jonge et al., 2013) and 7-methyl isomers (Ding et al., 2016) reacts to environment forcing : the 5-methyl correlates mainly with temperature (Naafs et al., 2017), while 6 and 7-methyl seem to react to moisture and pH changes (Yang et al., 2015, Ding et al., 2016)."(L. 69)

Line 265-270: this section needs a bit of re-writing (and maybe thinking). We know isoGDGTs are also produced in soils and peats, not just in lakes and not only in the water column. So that is not the reason that you see no clear correlation with climate parameters. See the discussion in (the supplementary information of) (Naafs et al., 2018) on the distribution of isoGDGTs in peat and the lack of correlation with temperature/pH. It is interesting that in these samples from dry environments crenarchaeol is abundant because the abundance of crenarchaeol in soils/peat has been inferred to indicate dry conditions (see for example Zheng et al., 2015), I suggest

the authors expand on this.

**Response:** We agree with this observation. We have tried the check the crenarcheol-MAP relationships, and we did not find any. The best (but weak) result (Figure 3 in the interactive comments)) found linked the ratio crenarcheol/regioisomer.

**Applied changes :** (L. 82)"*Iso-GDGT pattern in lake sediments do not really diverge from surface samples which leads to postulate that the in-situ production of iso-GDGTs in shallow lakes like MMNT5C12 is reduced (Fig. 4.A). Then, it appears that the iso-GDGT soil-produced are dominated by crenarcheol in accordance with studies on aridity impact (Zheng et al., 2015). However, no relationship exists between [crenarcheol] and MAP ($R^2$ = 0.14, $p - value$ > 0.005). The putative regio-isomer reaction linked to MAP (Buckles et al., 2016) is not evidenced in NMSDB.*"

Line 274: I don't understand this sentence

**Response and applied changes**: The text has been modified (L. 292): "*Particularly, the PC1 represents 22.77 % of the total variance and distinguishes two opposed poles: the 5-methyl group (mostly with PC1 >-0.3 associated with steppe-forest and forest sites) and the 6, 7-methyl groups on the far negative PC1 values associated with steppe and desert sites.*"

Figure 4: change legend to state "lake sediments, n=65"

**Response and applied changes**: Modified accordingly

Line 279: what kind of surface samples? Soil surface?

**Response and applied changes**: To emphasize the influence of sample type, the paragraph has been fully rewritten: "The sediment samples from the lake MMNT5C12, used as past sequence comparison, are more homogeneous than the soil surface samples, especially when compared with the moss polsters ( wide variability (Fig. 4B)). Generally, soil samples are more relevant analogues to lake sediments than moss polsters (especially [IIIa'], [IIa] and [Ia] relative abundances in Fig.4B). This variability shows an influence of the sample type on br–GDGT responses. On the other hand, sample type also bears climate and environment information, since soils and moss polsters originate mainly from steppe to desert environments and forest/alpine meadows, respectively."

Line 297-298: On what basis were these 9 selected?

**Response and applied changes**: The 15 models correspond to the best fitting model for each parameter added. Then we pick-up 9 models in the 15 whenever the addition of a parameter did not change significantly the statistical values ($R^2$, AIC and RMSE) more than a decimal of the previous model (with one parameter less). The sentence paragraph 4.2.3 have been modified to "*Within the 15 models (one model for each parameter addition), the 9 more contrasted ones were selected for discussion (Table 2)*."(L. 318)

Expand Section 5.3: I think the assumption for this section is flawed. Of course, a local calibration based on a certain set of surface samples will have a better correlation with the MAAT then a global calibration applied to these surface samples. You chose your local calibration using these same samples.

[Figure]

**Response and applied changes**: This observation is true for the cross-values made on samples also used for the calibration (the right part of all the panels A1, B1 and C1 on the fig.7) but it is not the case anymore for the "Cross-value" section of this same figure. Indeed, the 6 samples selected from the MMNT5C12 cores are all independent from our local br-GDGT calibration. The discussion of section 5.3 is based upon the panels of this graph. Since the discussion is based on an independent set of cross-value samples, the assumption should not be affected. We have tried to clarify by adding the following sentence: (L. 368) "*We tested both approaches on our datasets with a cross-value effectuated on the NMSDB-independent set of MMNT5C12 core samples*" at the beginning of part 5.3.

Line 379: also see discussion in (De Jonge et al., 2014; Naafs et al., 2017) on brGDGTs in dry soils

**Response and applied changes**: Some of the observations made on our data set corroborate the discussions in De Jonge et al., (2014) and Naafs et al., (2017), we have mentioned it in different parts of the manuscript:

- In the results (part 4.2.2, L. 304) "*The lower MAP match with 6 or 7-Me GDGTs, such as [IIIa'], [IIa'], [IIa"] in line with De Jonge et al., (2014b).*"

- In the discussion (part 5.4, L. 407) : "*First, the commonly used br–GDGT indexes (MBT and CBT) are not relevant for arid areas with $MAP < 500 mm.yr-1$ (Dirghangi et al., 2013; Menges et al., 2014) because of the relationships between low soil water content and soil br–GDGT preservation and conservation interfering with that of br–GDGT / climate parameters (Dang et al., 2016). The MAAT models built on MBT and MBT' indexes provide colder reconstructions (Fig. 7.C2) as it has been shown De Jonge et al. (2014b) because the arid soils favor 6-Methyl (by pH raising) and drive to the MBT flattening. This explain*

*the colder than real climate reconstructions provided by the* $\text{MAAT}_{\text{Ding}}$ *and* $\text{MAAT}_{\text{MBT}'-\text{DJ}}$ *calibrations."*